# Exploiting Tradeoffs for Exact Recovery in Heterogeneous Stochastic Block Models

**Amin Jalali**
Department of Electrical Engineering
University of Washington
Seattle, WA 98195
amjalali@uw.edu

**Qiyang Han**
Department of Statistics
University of Washington
Seattle, WA 98195
royhan@uw.edu

**Ioana Dumitriu**
Department of Mathematics
University of Washington
Seattle, WA 98195
dumitriu@uw.edu

**Maryam Fazel**
Department of Electrical Engineering
University of Washington
Seattle, WA 98195
mfazel@uw.edu

## Abstract

The Stochastic Block Model (SBM) is a widely used random graph model for networks with communities. Despite the recent burst of interest in community detection under the SBM from statistical and computational points of view, there are still gaps in understanding the fundamental limits of recovery. In this paper, we consider the SBM in its full generality, where there is no restriction on the number and sizes of communities or how they grow with the number of nodes, as well as on the connectivity probabilities inside or across communities. For such stochastic block models, we provide guarantees for exact recovery via a semidefinite program as well as upper and lower bounds on SBM parameters for exact recoverability. Our results exploit the tradeoffs among the various parameters of heterogenous SBM and provide recovery guarantees for many new interesting SBM configurations.

## 1 Introduction

A fundamental problem in network science and machine learning is to discover structures in large, complex networks (e.g., biological, social, or information networks). Community or cluster detection underlies many decision tasks, as a basic step that uses pairwise relations between data points in order to understand more global structures in the data. Applications include recommendation systems [27], image segmentation [24, 20], learning gene network structures in bioinformatics, e.g., in protein detection [9] and population genetics [17].

In spite of a long history of heuristic algorithms (see, e.g., [18] for an empirical overview), as well as strong research interest in recent years on the theoretical side as briefly reviewed in the sequel, there are still gaps in understanding the fundamental information theoretic limits of recoverability (i.e., if there is enough information to reveal the communities) and computational tractability (if there are efficient algorithms to recover them). This is particularly true in the case of sparse graphs (that test the limits of recoverability), graphs with heterogeneous communities (communities varying greatly in size and connectivity), graphs with a number of communities that grows with the number of nodes, and partially observed graphs (with various observation models).

## 1.1 Exact Recovery for Heterogenous Stochastic Block Model

The stochastic block model (SBM), first introduced and studied in mathematical sociology by Holland, Laskey and Leinhardt in 1983 [16], can be described as follows. Consider $n$ vertices partitioned into $r$ *communities* $V_1, V_2, \ldots, V_r$, of sizes $n_1, n_2, \ldots, n_r$. We endow the $k$th community with an Erdős-Rényi random graph model $\mathcal{G}(n_k, p_k)$ and draw an edge between pairs of nodes in different communities independently with probability $q$; i.e., for any pair of nodes $i$ and $j$, if $i, j \in V_k$ for some $k \in \{1, \ldots, r\}$ we draw an edge with probability $p_k$, and draw an edge with probability $q$ if they are in different communities. We assume $q < \min_k p_k$ in order for the idea of communities to make sense. This defines a distribution over random graphs known as the stochastic block model. In this paper, we assume the above model while allowing the number of communities to grow with the number of nodes (similar to [13, 15, 23]). We refer to this model as the *heterogeneous stochastic block model* to contrast our study of this general setting with previous works on special cases of SBM such as 1) homogenous SBM where the communities are *equivalent* (they are of the same size and the connectivity probabilities are equal,) e.g., [12], or, 2) SBM with linear-sized communities, where the number of communities is *fixed* and all community sizes are $O(n)$; e.g., [1].

## 1.2 Statistical and Computational Regimes

What we can infer about the community structure from a single draw of the random graph varies based on the regime of model parameters. Often, the following scenarios are considered.

1. *Recovery,* where the proportion of misclassified nodes is negligible; either 0 (corresponding to exact recovery with *strong consistency*, and considered in [12, 1]) or asymptotically 0 (corresponding to exact recovery with *weak consistency* as considered in [23, 22, 28]) as the number of nodes grows.
2. *Approximation,* where a finite fraction (bounded away from 1) of the vertices is recovered. This regime was first introduced in [13, 14], and has been considered in many other works since then; e.g., see [15] and references therein.

Both recovery and approximation can be studied from statistical and computational points of view.

*Statistically,* one can ask about the parameter regimes for which the model can be recovered or approximated. Such characterizations are specially important when an information-theoretical lower bound (below which recovery is not possible with high probability) is shown to be achievable with an algorithm (with high probability), hence characterizing a *phase transition* in model parameters. Recently, there has been significant interest in identifying such *sharp thresholds* for various parameter regimes.

*Computationally,* one might be interested to study algorithms for recovery or approximation. In the older approach, algorithms were studied to provide upper bounds on the parameter regimes for recovery or approximation. See [10] or [1, Section 5] for a summary of such results. More recently, the paradigm has shifted towards understanding the limitations and strengths of tractable methods (e.g. see [21] on semidefinite programming based methods) and assessing whether successful retrieval can be achieved by tractable algorithms at the sharp statistical thresholds or there is a *gap*. So far, it is understood that there is no such gap in the case of exact recovery (weak and strong) and approximation of binary SBM as well as the exact recovery of linear-sized communities [1]. However, this is still an open question for more general cases; e.g., see [2] and the list of unresolved conjectures therein.

The statistical-computational picture for SBM with only two equivalent communities has been fully characterized in a series of recent papers. Apart from the binary SBM, the best understood cases are where there is a finite number $r$ of equivalent or linear-sized communities. Outside of the settings described above, the full picture has not yet emerged and many questions are unresolved.

## 1.3 This paper

The community detection problem studied in this paper is stated as: given the adjacency matrix of a graph generated by the heterogenous stochastic block model, for what SBM parameters we can recover the labels of *all* vertices, with high probability, using an algorithm that has been proved to do so. We consider a convex program in (2.4) and an estimator similar to the maximum likelihood

estimator in (2.5) and characterize parts of the model space for which exact recovery is possible via these algorithms. Theorems 1 and 2 provide sufficient conditions for the convex recovery program and Theorem 3 provides sufficient conditions for the modified maximum likelihood estimator to exactly recover the underlying model. In Section 2.3, we extend the above bounds to the case of partial observations, i.e., when each entry of the matrix is observed uniformly at random with some probability $\gamma$ and the results are recorded. We also provide an information-theoretic lower bound, describing an impossibility regime for exact recovery in heterogenous SBM in Theorem 4. All of our results only hold with high probability, as this is the best one can hope for; with tiny probability the model can generate graphs like the complete graph where the partition is unrecoverable.

The results of this paper provide a clear improvement in the understanding of stochastic block models by exploiting tradeoffs among SBM parameters. We identify a key parameter (or summary statistic), defined in (2.1) and referred to as *relative density*, which shows up in our results and provides improvements in the statistical assessment and efficient computational approaches for certain configurations of heterogenous SBM; examples are given in in Section 3 to illustrate a number of such beneficial tradeoffs such as

- semidefinite programming can successfully recover communities of size $O(\sqrt{\log n})$ under mild conditions on other communities (see Example 3 for details) while $\log n$ has long been believed to be the threshold for the smallest community size.
- The sizes of the communities can be very spread, or the inter- and intra-community probabilities can be very close, and the model still be efficiently recoverable, while existing methods (e.g., *peeling strategy* [3]) providing false negatives.

While these results are a step towards understanding the information-computational picture about the heterogenous SBM with a growing number of communities, we cannot comment on phase transitions or a possible information-computational gap (see Section 1.2) in this setup based on the results of this paper.

## 2   Main Results

Consider the heterogenous stochastic block model described above. In the proofs, we can allow for isolated nodes (communities of size 1) which are omitted from the model here to simplify the presentation. Denote by $\mathcal{Y}$ the set of admissible adjacency matrices according to a community assignment as above, i.e.,

$$\mathcal{Y} := \{Y \in \{0,1\}^{n \times n} : Y \text{ is a valid community matrix w.r.t. } V_1, \ldots, V_r \text{ where } |V_k| = n_k\}.$$

Define the *relative density of community $k$* as

$$\rho_k = (p_k - q)n_k \tag{2.1}$$

which can be seen as the increase in the average degree of a node in community $k$ in the SBM, relative to its average degree in an Erdős-Rényi model. Define $n_{\min}$ and $n_{\max}$ as the minimum and maximum of $n_1, \ldots, n_k$ respectively. The total variance over the $k$th community is defined as $\sigma_k^2 = n_k p_k (1 - p_k)$, and we let $\sigma_0^2 = nq(1 - q)$. Moreover, consider

$$\sigma_{\max}^2 = \max_{k=1,\ldots,r} \sigma_k^2 = \max_{k=1,\ldots,r} n_k p_k (1 - p_k). \tag{2.2}$$

A Bernoulli random variable with parameter $p$ is denoted by $\mathrm{Ber}(p)$, and a Binomial random variable with parameters $n$ and $p$ is denoted by $\mathrm{Bin}(n, p)$. The Neyman Chi-square divergence between the two discrete random variables $\mathrm{Ber}(p)$ and $\mathrm{Ber}(q)$ is given by

$$\widetilde{D}(p, q) := \frac{(p - q)^2}{q(1 - q)} \tag{2.3}$$

and we have $\widetilde{D}(p, q) \geq D_{\mathrm{KL}}(p, q) := D_{\mathrm{KL}}(\mathrm{Ber}(p), \mathrm{Ber}(q))$. Chi-square divergence is an instance of a more general family of divergence functions called $f$-divergences or Ali-Silvey distances. This family also has KL-divergence, total variation distance, Hellinger distance and Chernoff distance as special cases. Moreover, the divergence used in [1] is an $f$-divergence.

Lastly, $\log$ denotes the natural logarithm (base $e$), and the notation $\theta \gtrsim 1$ is equivalent to $\theta \geq O(1)$.

## 2.1 Convex Recovery

Inspired by the success of semidefinite programs in community detection (e.g., see [15, 21]) we consider a natural convex relaxation of the maximum likelihood estimator, similar to the one used in [12], for exact recovery of the heterogeneous SBM with a growing number of communities. Assuming that $\zeta = \sum_{k=1}^{r} n_k^2$ is known, we solve

$$
\begin{aligned}
\hat{Y} \;=\; & \underset{Y}{\arg\max} && \textstyle\sum_{i,j} A_{ij} Y_{ij} \\
& \text{subject to} && \|Y\|_\star \le n \;,\; \textstyle\sum_{i,j} Y_{ij} = \zeta \;,\; 0 \le Y_{ij} \le 1 \,.
\end{aligned}
\tag{2.4}
$$

where $\|\cdot\|_\star$ denotes the nuclear norm (the sum of singular values of the matrix).

We prove two theorems giving conditions under which the above convex program outputs the true community matrix with high probability. In establishing these performance guarantees, we follow the standard *dual certificate* argument in convex analysis while utilizing strong matrix concentration results from random matrix theory [8, 25, 26, 5]. These results allow us to bound the spectral radius of the matrix $A - \mathbb{E}[A]$ where $A$ is an instance of adjacency matrix generated under heterogenous SBM. The proofs for both theorems along with the matrix concentration bounds are given in Appendix A.

**Theorem 1** *Under the heterogenous stochastic block model, the output of the semidefinite program in* (2.4) *coincides with $Y^\star$ with high probability, provided that*

$$
\rho_k^2 \gtrsim \sigma_k^2 \log n_k \;,\quad \widetilde{D}(p_{\min}, q) \gtrsim \frac{\log n_{\min}}{n_{\min}} \;,\quad \rho_{\min}^2 \gtrsim \max\{\sigma_{\max}^2,\, nq(1-q),\, \log n\}
$$

*and $\sum_{k=1}^{r} n_k^{-\alpha} = o(1)$ for some $\alpha > 0$.*

*Proof Sketch.* For $Y^\star$ to be the unique solution of (2.4), we need to show that for any feasible $Y \ne Y^\star$, the following quantity

$$
\langle A, Y^\star - Y \rangle = \langle \mathbb{E}[A], Y^\star - Y \rangle + \langle A - \mathbb{E}[A], Y^\star - Y \rangle
$$

is strictly positive. In bounding the second term above, we make use of the constraint $\|Y\|_\star \le n = \|Y^\star\|_\star$ by constructing a *dual certificate* from $A - \mathbb{E}[A]$. This is where the bounds on the spectral norm (dual norm for the nuclear norm) of $A - \mathbb{E}[A]$ enter and we use matrix concentration bounds (see Lemma 7 in Appendix A).

The first condition of Theorem 1 is equivalent to each community being connected, second condition ensures that each community is identifiable ($p_{\min} - q$ is large enough), and the third condition requires minimal density to dominate global variability. The assumption $\sum_{k=1}^{r} n_k^{-\alpha} = o(1)$ is tantamount to saying that the number of tiny communities cannot be too large (e.g., the number of polylogarithmic-size communities cannot be a power of $n$). In other words, one needs to have mostly large communities (growing like $n^\epsilon$, for some $\epsilon > 0$) for this assumption to be satisfied. Note, however, that the condition does *not* restrict the number of communities of size $n^\epsilon$ for any fixed $\epsilon > 0$. In fact, Theorem 1 allows us to describe a regime in which *tiny* communities of size $O(\sqrt{\log n})$ are recoverable provided that they are very dense and that only few tiny or small communities exist; see Example 3. The second theorem imposes more stringent conditions on the relative density, hence only allowing for communities of size down to $\log n$, but relaxes the condition that only a small number of nodes can be in small communities.

**Theorem 2** *Under the heterogenous stochastic block model, the output of the semidefinite program in* (2.4) *coincides with $Y^\star$, with high probability, provided that*

$$
\rho_k^2 \gtrsim \sigma_k^2 \log n \;,\quad \widetilde{D}(p_{\min}, q) \gtrsim \frac{\log n}{n_{\min}} \;,\quad \rho_{\min}^2 \gtrsim \max\{\sigma_{\max}^2,\, nq(1-q)\} \,.
$$

The proof of Theorem 2 is similar to the proof of Theorem 1 except that we use a different matrix concentration bound (see Lemma 10 in Appendix A).

## 2.2 Recoverability Lower and Upper Bounds

Next, we consider an estimator, inspired by maximum likelihood estimation, and identify a subset of the model space which is exactly recoverable via this estimator. The proposed estimation approach

is not computationally tractable and is only used to examine the conditions for which exact recovery is possible. For a fixed $Y \in \mathcal{Y}$ and an observed matrix $A$, the likelihood function is given by

$$\mathbb{P}_Y(A) = \prod_{i<j} p_{\tau(i,j)}^{A_{ij}Y_{ij}} (1 - p_{\tau(i,j)})^{(1-A_{ij})Y_{ij}} q^{A_{ij}(1-Y_{ij})} (1-q)^{(1-A_{ij})(1-Y_{ij})},$$

where $\tau : \{1, \ldots, n\}^2 \to \{1, \ldots, r\}$ and $\tau(i,j) = k$ if and only if $i, j \in V_k$, and arbitrary in $\{1, \ldots, r\}$ otherwise. The log-likelihood function is given by

$$\log \mathbb{P}_Y(A) = \sum_{i<j} \log \frac{(1-q)p_{\tau(i,j)}}{q(1-p_{\tau(i,j)})} A_{ij} Y_{ij} + \sum_{i<j} \log \frac{1 - p_{\tau(i,j)}}{1-q} Y_{ij} + \text{ terms not involving } \{Y_{ij}\}.$$

Maximizing the log-likelihood involves maximizing a weighted sum of $\{Y_{ij}\}$'s where the weights depend on the (usually unknown) values of $q, p_1, \ldots, p_r$. To be able to work with less information, we will use the following modification of maximum likelihood estimation, which only uses the knowledge of $n_1, \ldots, n_r$,

$$\hat{Y} = \arg\max_{Y \in \mathcal{Y}} \sum_{i,j=1}^{n} A_{ij} Y_{ij}. \tag{2.5}$$

**Theorem 3** *Suppose $n_{\min} \geq 2$ and $n \geq 8$. Under the heterogenous stochastic block model, if*

$$\rho_{\min} \geq 4(17 + \eta) \left( \frac{1}{3} + \frac{p_{\min}(1 - p_{\min}) + q(1-q)}{p_{\min} - q} \right) \log n,$$

*for some choice of $\eta > 0$, then the optimal solution $\hat{Y}$ of the non-convex recovery program in (2.5) coincides with $Y^\star$, with a probability not less than $1 - 7 \frac{p_{\max} - q}{p_{\min} - q} n^{2 - \eta}$.*

Notice that $\rho_{\min} = \min_{k=1,\ldots,r} n_k(p_k - q)$ and $p_{\min} = \min_{k=1,\ldots,r} p_k$ do not necessarily correspond to the same community. Similar to the proof of Theorem 1, we establish $\langle A, Y^\star - Y \rangle > 0$ for any $Y \in \mathcal{Y}$, while this time, we use a counting argument (see Lemma 11 in Appendix B) similar to the one in [12]. The proofs for this Theorem and the next one are given in Appendix B.

Finally, to provide a better picture of community detection for heterogenous SBM we provide the following necessary conditions for exact recovery. Notice that Theorems 1 and 2 require $\widetilde{D}(q, p_k)$ (in their first condition) and $\widetilde{D}(p_k, q)$ (in their second condition) to be bounded from below for recoverability by the SDP. Similarly, the conditions of Theorem 4 can be seen as average-case and worst-case upper bounds on these divergences.

**Theorem 4** *If any of the following conditions holds,*

*(1) $2 \leq n_k \leq n/e$, and $4 \sum_{k=1}^{r} n_k^2 \widetilde{D}(p_k, q) \leq \frac{1}{2} \sum_k n_k \log \frac{n}{n_k} - r - 2$*

*(2) $n \geq 128$, $r \geq 2$ and $\max_k \left\{ n_k \widetilde{D}(p_k, q) + n_k \widetilde{D}(q, p_k) \right\} \leq \frac{1}{12} \log(n - n_{\min})$*

*then $\inf_{\hat{Y}} \sup_{Y^\star \in \mathcal{Y}} \mathbb{P}[\hat{Y} \neq Y^\star] \geq \frac{1}{2}$ where the infimum is taken over all measurable estimators $\hat{Y}$ based on the realization $A$ generated according to the heterogenous stochastic block model.*

## 2.3 Partial Observations

In the general stochastic block model, we assume that the entries of a symmetric adjacency matrix $A \in \{0,1\}^{n \times n}$ have been generated according to a combination of Erdős-Rényi models with parameters that depend on the true community matrix. In the case of partial observations, we assume that the entries of $A$ has been observed independently with probability $\gamma$. In fact, every entry of the input matrix falls into one of these categories: *observed as one* denoted by $\Omega_1$, *observed as zero* denoted by $\Omega_0$, and *unobserved* which corresponds to $\Omega^c$ where $\Omega = \Omega_0 \cup \Omega_1$. If an estimator only takes the observed part of the matrix as the input, one can revise the underlying probabilistic model to incorporate both the stochastic block model and the observation model; i.e. a revised distribution for entries of $A$ as

$$A_{ij} = \begin{cases} \text{Ber}(\gamma p_k) & i, j \in V_k \text{ for some } k \\ \text{Ber}(\gamma q) & i \in V_k \text{ and } j \in V_l \text{ for } k \neq l. \end{cases}$$

yields the same output from an estimator that only takes in the observed values. Therefore, the estimators in (2.4) and (2.5), as well as the results of Theorems 1, 2, 3, can be easily adapted to the case of partially observed graphs. It is worth mentioning that the above model for partially observed SBM (which is another SBM) is different from another random model known as Censored Block Model (CBM) [4]. In SBM, absence of an edge provides information, whereas in CBM it does not.

## 3 Tradeoffs in Heterogenous SBM

As it can be seen from the results presented in this paper, and the main summary statistics they utilize (the relative densities $\rho_1, \ldots, \rho_r$), the parameters of SBM can vary significantly and still satisfy the same recoverability conditions. In the following, we examine a number of such tradeoffs which leads to recovery guarantees for interesting SBM configurations. Here, a *configuration* is a list of community sizes $n_k$, their connectivity probabilities $p_k$, and the inter-community connectivity probability $q$. A triple $(m, p, k)$ represents $k$ communities of size $m$ each, with connectivity parameter $p$. We do not worry about whether $m$ and $k$ are always integers; if they are not, one can always round up or down as needed so that the total number of vertices is $n$, without changing the asymptotics. Moreover, when the $O(\cdot)$ notation is used, we mean that appropriate constants can be determined. A detailed list of computations for the examples in this section are given in Appendix D.

**Table 1:** A summary of examples in Section 3. Each row gives the important aspect of the corresponding example as well as whether, under appropriate regimes of parameters, it would satisfy the conditions of the theorems proved in this paper.

|  | importance | convex recovery by Thm. 1 | convex recovery by Thm. 2 | recoverability by Thm. 3 |
|---|---|---|---|---|
| Ex. 1 | $\{\rho_k\}$ instead of $(p_{\min}, n_{\min})$ | × | × | ✓ |
| Ex. 2 | stronger guarantees for convex recovery | ✓ | ✓ | ✓ |
| Ex. 3 | $n_{\min} = \sqrt{\log n}$ | ✓ | × | × |
| Ex. 4 | many small communities, $n_{\max} = O(n)$ | ✓ | ✓ | ✓ |
| Ex. 5 | $n_{\min} = O(\log n)$, spread in sizes | × | ✓ | ✓ |
| Ex. 6 | small $p_{\min} - q$ | ✓ | ✓ | ✓ |

*Better Summary Statistics.* It is intuitive that using summary statistics such as $(p_{\min}, n_{\min})$, for a heterogenous SBM where $n_k$'s and $p_k$'s are allowed to take very different values, can be very limiting. Examples 1 and 2 are intended to give configurations that are guaranteed to be recoverable by our results but fail the existing recoverability conditions in the literature.

**Example 1** Suppose we have two communities of sizes $n_1 = n - \sqrt{n}$, $n_2 = \sqrt{n}$, with $p_1 = n^{-2/3}$ and $p_2 = 1/\log n$ while $q = n^{-2/3-0.01}$. The bound we obtain here in Theorem 3 makes it clear that this case is theoretically solvable (the modified maximum likelihood estimator successfully recovers it). By contrast, Theorem 3.1 in [7] (specialized for the case of no outliers), requiring

$$n_{\min}^2 (p_{\min} - q)^2 \gtrsim (\sqrt{p_{\min} n_{\min}} + \sqrt{nq})^2 \log n, \tag{3.1}$$

would fail and provide no guarantee for recoverability.

**Example 2** Consider a configuration as

$$(n - n^{2/3}, n^{-1/3+\epsilon}, 1) \ , \ (\sqrt{n}, O(\tfrac{1}{\log n}), n^{1/6}) \ , \ q = n^{-2/3+3\epsilon}$$

where $\epsilon$ is a small quantity, e.g., $\epsilon = 0.1$. Either of Theorems 1 and 2 certify this case as recoverable via the semidefinite program (2.4) with high probability. By contrast, using the $p_{\min} = n^{-1/3+\epsilon}$ and $n_{\min} = \sqrt{n}$ heuristic, neither the condition of Theorem 3.1 in [7] (given in (3.1)) nor the condition of Theorem 2.5 in [12] is fulfilled, hence providing no recovery guarantee for this configuration.

### 3.1 Small communities can be efficiently recovered

Most algorithms for clustering the SBM run into the problem of small communities [11, 6, 19], often because the models employed do not allow for enough parameter variation to identify the key quantities involved. The next three examples attempt to provide an idea of how small the community

sizes can be, how many small communities are allowed, and how wide the spread of community sizes can be, as characterized by our results.

**Example 3 (smallest community size for convex recovery)** Consider a configuration as

$$(\sqrt{\log n}, O(1), m) \ , \ (n_2, O(\tfrac{\log n}{\sqrt{n}}), \sqrt{n}) \ , \ q = O(\tfrac{\log n}{n})$$

where $n_2 = \sqrt{n} - m\sqrt{\log n/n}$ to ensure a total of $n$ vertices. Here, we assume $m \leq n/(2\sqrt{\log n})$ which implies $n_2 \geq \sqrt{n}/2$. It is straightforward to verify the conditions of Theorem 1.

To our knowledge, *this is the first example in the literature for which semidefinite programming based recovery works and allows the recovery of (a few) communities of size smaller than* $\log n$. Previously, $\log n$ was considered to be the standard bound on the community size for exact recovery, as illustrated by Theorem 2.5 of [12] in the case of equivalent communities. We have thus shown that it is possible, in the right circumstances (when sizes are spread and the smaller the community the denser it is), to recover very small communities (up to $\sqrt{\log n}$ size), *if there are just a few of them (at most polylogarithmic in* $n$). The significant improvement we made in the bound on the size of the smallest community is due to the fact that we were able to perform a closer analysis of the semidefinite program by utilizing stronger matrix concentration bounds, mainly borrowed from [8, 25, 26, 5]. For more details, see Appendix A.2.

Notice that the condition of Theorem 3 is *not* satisfied. This is not an inconsistency (as Theorem 3 gives only an upper bound for the threshold), but indicates the limitation of this theorem in characterizing all recoverable cases.

*Spreading the sizes.* As mentioned before, while Theorem 1 allows for going lower than the standard $\log n$ bound on the community size for exact recovery, it requires the number of very small communities to be relatively small. On the other hand, Theorem 2 provides us with the option of having many small communities but requires the smallest community to be of size $O(\log n)$. We explore two cases with many small communities in the following.

**Example 4** Consider a configuration where small communities are dense and there is one big community,

$$(\tfrac{1}{2}n^\epsilon, O(1), n^{1-\epsilon}) \ , \ (\tfrac{1}{2}n, n^{-\alpha}\log n, 1) \ , \ q = O(n^{-\beta}\log n)$$

with $0 < \epsilon < 1$ and $0 < \alpha < \beta < 1$. We are interested to see how large the number of small communities can be. Then the conditions of Theorems 1 and 2 both require that

$$\tfrac{1}{2}(1-\alpha) < \epsilon < 2(1-\alpha) \ , \ \ \epsilon > 2\alpha - \beta \tag{3.2}$$

and are depicted in Figure 1. Since we have not specified the constants in our results, we only consider strict inequalities.

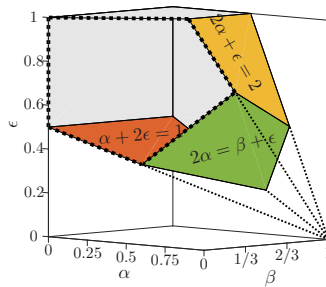

**Figure 1:** The space of parameters in Equation (3.2). The face defined by $\beta = \alpha$ is shown with dotted edges. The three gray faces in the back correspond to $\beta = 1$, $\alpha = 0$ and $\epsilon = 1$. The green plane (corresponding to the last condition in (3.2)) comes from controlling the intra-community interactions uniformly (interested reader is referred to Equations (A.8) and (A.9) in the supplement material) which might be only an artifact of our proof and can be possibly improved.

Notice that the small communities are as dense as can be, but the large one is not necessarily very dense. By picking $\epsilon$ to be just over $1/4$, we can make $\alpha$ just shy of $1/2$, and $\beta$ very close to $1$. As

far as we can tell, there are no results in the literature surveyed that cover such a case, although the clever "peeling" strategy introduced in [3] would recover the largest community. The strongest result in [3] that seems applicable here is Corollary 4 (which works for non-constant probabilities). The algorithm in [3] works to recover a large community (larger than $O(\sqrt{n}\log^2 n)$), subject to existence of a gap in the community sizes (roughly, there should be no community sizes between $O(\sqrt{n})$ and $O(\sqrt{n}\log^2 n)$). Therefore, in this example, after a single iteration, the algorithm will stop, despite the continued existence of a gap, as there is no community with size above the gap. Hence the "peeling" strategy on this example would fail to recover all the communities.

**Example 5** Consider a configuration with many small dense communities of size $\log n$. We are interested to see how large the spread of community sizes can be for the semidefinite program to work. As required by Theorems 1 and 2 and to control $\sigma_{\max}$ (defined in (2.2)), the larger a community the smaller its connectivity probability should be; therefore we choose the largest community at the threshold of connectivity (required for recovery). Consider the community sizes and probabilities:

$$(\log n,\, O(1),\, n/\log n - m\sqrt{n/\log n})\ ,\ (\sqrt{n\log n},\, O(\sqrt{(\log n)/n}),\, m)\ ,\ q = O((\log n)/n)$$

where $m$ is a constant. Again, we round up or down where necessary to make sure the sizes are integers and the total number of vertices is $n$. All the conditions of Theorem 2 are satisfied and exact convex recovery is possible via the semidefinite program. Note that the last condition of Theorem 1 is not satisfied since there are too many small communities. Also note that alternative methods proposed in the literature surveyed would not be applicable; in particular, the gap condition in [3] is not satisfied for this case from the start.

## 3.2 Weak communities are efficiently recoverable

The following examples illustrate how small $p_{\min} - q$ can be in order for the recovery, respectively, the convex recovery algorithms to still be guaranteed to work. When some $p_k$ is very close to $q$, the Erdős-Rényi model $\mathcal{G}(n_k, p_k)$ looks very similar to the ambient edges from $\mathcal{G}(n, q)$. Again, we are going to exploit the possible tradeoffs in the parameters of SBM to guarantee recovery. Note that the difference in $p_{\min} - q$ for the two types of recovery is noticeable, indicating that there is a significant difference between what we know to be recoverable and what we can recover efficiently by our convex method. We consider both dense graphs (where $p_{\min}$ is $O(1)$) and sparse ones.

**Example 6** Consider a configuration where all of the probabilities are $O(1)$ and

$$(n_1,\, p_{\min},\, 1)\ ,\ (n_{\min},\, p_2,\, 1)\ ,\ (n_3,\, p_3,\, \tfrac{n-n_1-n_{\min}}{n_3})\ ,\ q = O(1)$$

where $p_2 - q$ and $p_3 - q$ are $O(1)$. On the other hand, we assume $p_{\min} - q = f(n)$ is small. For recoverability by Theorem 3, we need $f(n) \gtrsim (\log n)/n_{\min}$ and $f^2(n) \gtrsim (\log n)/n_1$. Notice that, since $n \gtrsim n_1 \gtrsim n_{\min}$, we should have $f(n) \gtrsim \sqrt{\log n/n}$. For the convex program to recover this configuration (by Theorem 1 or 2), we need $n_{\min} \gtrsim \sqrt{n}$ and $f^2(n) \gtrsim \max\{n/n_1^2,\, \log n/n_{\min}\}$, while all the probabilities are $O(1)$.

Note that if all the probabilities, as well as $p_{\min} - q$, are $O(1)$, then by Theorem 3 all communities down to a logarithmic size should be recoverable. However, the success of convex recovery is guaranteed by Theorems 1 and 2 when $n_{\min} \gtrsim \sqrt{n}$.

For a similar configuration to Example 6, where the probabilities are not $O(1)$, recoverability by Theorem 3 requires $f(n) \gtrsim \max\{\sqrt{p_{\min}(\log n)/n},\, n^{-c}\}$ for some appropriate $c > 0$.

# 4  Discussion

We have provided a series of extensions to prior works (especially [12, 1]) by considering the exact recovery for stochastic block model in its full generality with a growing number of communities. By capturing the tradeoffs among the various parameters of SBM, we have identified interesting SBM configurations that are efficiently recoverable via semidefinite programs. However there are still interesting problems that remain open. Sharp thresholds for recovery or approximation of heterogeneous SBM, models for partial observation (non-uniform, based on prior information, or adaptive as in [28]), as well as overlapping communities (e.g., [1]) are important future directions. Moreover, other estimators similar to the ones considered in this paper can be analyzed; e.g. when the unknown parameters in the maximum likelihood estimator, or $\zeta$ in (2.4), are estimated from the given observations.

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
