[Supplementary Material · supplement_material.pdf]

# Exploiting Tradeoffs for Exact Recovery in Heterogeneous Stochastic Block Models Supplement Material

## A Proofs for Convex Recovery

In the following, we present the proofs of Theorems 1 and 2. The matrix concentration bounds play an important role in these proofs, and are given as Lemmas 7 and 10.

### A.1 Notation

In this paper, we consider the heterogenous stochastic block model described in Section 1.1. Consider a partition of the $n$ nodes into $V_0, V_1, \ldots, V_r$, where $|V_k| = n_k$, $k = 0, 1, \ldots, r$. Consider $\bar{n} = \sum_{k=1}^{r} n_k$ and denote the number of isolated nodes by $n_0$; hence, $n_0 + \bar{n} = n$. Ignoring $n_0$, we further define $n_{\min}$ and $n_{\max}$ as the minimum and maximum of $n_1, \ldots, n_r$ respectively. The nodes in $V_0$ are isolated and the nodes in $V_k$ form the community $\mathcal{C}_k = V_k \times V_k$, for $k = 1, \ldots, r$. The union of communities is denoted by $\mathcal{C} = \cup_{k=1}^{r} \mathcal{C}_k$ and $\mathcal{C}^c$ denotes the complement; i.e. $\mathcal{C}^c = \{(i, j) : (i, j) \notin \mathcal{C}_k \text{ for any } k = 1, \ldots, r, \text{ and } i, j = 1, \ldots, n\}$. Denote by $\mathcal{Y}$ the set of admissible adjacency matrices according to a community assignment as above, i.e.

$$\mathcal{Y} := \{Y \in \{0, 1\}^{n \times n} : Y \text{ is a valid community matrix w.r.t. } V_0, V_1, \ldots, V_r \text{ where } |V_k| = n_k\}.$$

We will denote by $\mathbf{1}_C \in \mathbb{R}^{n \times n}$ a matrix which is 1 on $C \subset \{1, \ldots, n\}^2$ and zero elsewhere. $\log$ denotes the natural logarithm (base $e$), and the notation $\theta \gtrsim 1$ is equivalent to $\theta \geq O(1)$. A Bernoulli random variable with parameter $p$ is denoted by $\mathrm{Ber}(p)$, and a Binomial random variable with parameters $n$ and $p$ is denoted by $\mathrm{Bin}(n, p)$. $\|\cdot\|_\star$ denotes the matrix nuclear norm or trace norm, i.e., the sum of singular values of the matrix. The dual to the nuclear norm is the spectral norm, denoted by $\|\cdot\|$.

Given a single graph drawn from the heterogenous stochastic block model, the goal is to recover the underlying community matrix $Y^\star \in \mathcal{Y}$ exactly. We will need the following definitions:

Define the *relative density of a community* as

$$\rho_k = (p_k - q)n_k$$

which gives $\sum_{k=1}^{r} \rho_k = \sum_{k=1}^{r} p_k n_k - qn$. Define the total variance $\sigma_k^2 = n_k p_k (1 - p_k)$ over the $k$th community, and let $\sigma_0^2 = nq(1 - q)$. Also, define

$$\sigma_{\max}^2 = \max_{k=1,\ldots,r} \sigma_k^2 = \max_{k=1,\ldots,r} n_k p_k (1 - p_k).$$

The Neyman Chi-square divergence (e.g., see [17]) between the two discrete random variables $\mathrm{Ber}(p)$ and $\mathrm{Ber}(q)$ is given by

$$\widetilde{D}(p, q) := \frac{(p - q)^2}{q(1 - q)}$$

and we have $\widetilde{D}(p, q) \geq D_{\mathrm{KL}}(p, q) := D_{\mathrm{KL}}(\mathrm{Ber}(p), \mathrm{Ber}(q))$; see (B.19). Chi-square divergence is an instance of a more general family of divergence functions called $f$-divergences or Ali-Silvey distances [6]. This family also has KL-divergence, total variation distance, Hellinger distance and Chernoff distance as special cases. Moreover, the divergence used in [3] is an $f$-divergence.

### A.2 Proof of Theorem 1

We are going to prove that under the heterogenous stochastic block model (HSBM), with high probability, the output of the convex recovery program in (2.4) coincides with the underlying community matrix $Y^\star = \sum_{k=1}^{r} \mathbf{1}_{V_k} \mathbf{1}_{V_k}^T$ provided that

$$\rho_k^2 \gtrsim n_k p_k (1 - p_k) \log n_k$$

$$(p_{\min} - q)^2 \gtrsim q(1 - q) \frac{\log n_{\min}}{n_{\min}} \tag{A.1}$$

$$\rho_{\min}^2 \gtrsim \max\left\{ \max_k n_k p_k (1 - p_k), \, nq(1 - q), \, \log n \right\}$$

as well as $\sum_{k=1}^r n_k^{-\alpha} = o(1)$ for some $\alpha > 0$.

Notice that $p_k(1 - p_k)n_k \gtrsim \log n_k$, for all $k = 1, \ldots, r$, is implied by the first condition, as mentioned in Remark 1.

**Remark 1** *For exact recovery to be possible, we need all communities (but at most one) to be connected. Therefore, in each subgraph, which is generated by $\mathcal{G}(n_k, p_k)$, we need $p_k n_k > \log n_k$, for $k = 1, \ldots, r$. Observe that this connectivity requirement is implicit in the first condition of Theorems 1, 2: for example, the first condition of Theorem 1 can be equivalently expressed as $n_k \widetilde{D}(q, p_k) \gtrsim \log n_k$. Moreover, for $q < p$, when both $p$ and $q/p$ are bounded away from 1, we have*

$$\widetilde{D}(q, p) = p\frac{(1 - q/p)^2}{1 - p} \approx p.$$

Before proving Theorem 1, we state a crucial result from random matrix theory that allows us to bound the spectral radius of the matrix $A - \mathbb{E}(A)$ where $A$ is an instance of adjacency matrix under HSBM. This result appears, for example, as Theorem 3.4 in [11][1]. Although Lemma 2 from [36] appears to state a weaker version of this result, the proof presented there actually supports the version we give below in Lemma 5. Finally, Lemma 8 from [38] states the same result and presents a very brief sketch of the proof idea, along the lines of the proof presented fully in [36].

**Lemma 5** *Let $A = \{a_{ij}\}$ be a $n \times n$ symmetric random matrix such that each $a_{ij}$ represents an independent random Bernoulli variable with $\mathbb{E}(a_{ij}) = p_{ij}$. Assume that there exists a constant $C_0$ such that $\sigma^2 = \max_{i,j} p_{ij}(1 - p_{ij}) \geq C_0 \log n/n$. Then for each constant $C_1 > 0$ there exists $C_2 > 0$ such that*

$$\mathbb{P}\left(\|A - \mathbb{E}(A)\| \geq C_2 \sigma \sqrt{n}\right) \leq n^{-C_1}.$$

As an immediate consequence of this, we have the following corollary.

**Corollary 6** *Let $A = \{a_{ij}\}$ be a $n \times n$ symmetric random matrix such that each $a_{ij}$ represents an independent random Bernoulli variable with $\mathbb{E}(a_{ij}) = p_{ij}$. Assume that there exists a constant $C_0$ such that $\sigma^2 = \max_{i,j} p_{ij}(1 - p_{ij}) \leq C_0 \log n/n$. Then for each constant $C_1 > 0$ there exists $C_3 > 0$ such that such that*

$$\mathbb{P}\left(\|A - \mathbb{E}(A)\| \geq C_3 \sqrt{\log n}\right) \leq n^{-C_1}.$$

*Proof.* The corollary follows from Lemma 5, by replacing the $(1, 1)$ entry of $A$ with a Bernoulli variable of probability $p_{11} = C_0 \log n/n$. Given that the old $(1, 1)$ entry and the new $(1, 1)$ entry are both Bernoulli variables, this can change $\|A - \mathbb{E}(A)\|$ by at most 1. The new maximal variance is equal to $\max_{i,j} p_{ij}(1 - p_{ij}) = C_0 \log n/n$. Therefore Lemma 5 is applicable to the new matrix and the conclusion holds. ∎

We use Lemma 5 to prove the following result.

**Lemma 7** *Let $A$ be generated according to the heterogenous stochastic block model (HSBM). Suppose*

*(1) $p_k(1 - p_k)n_k \gtrsim \log n_k$, for $k = 1, \ldots, r$, and*

*(2) there exists an $\alpha > 0$ such that $\sum_{k=0}^r n_k^{-\alpha} = o(1)$.*

*Then with probability at least $1 - o(1)$ we have*

$$\|A - \mathbb{E}(A)\| \lesssim \max_i \sqrt{p_i(1 - p_i)n_i} + \sqrt{\max\{q(1 - q)n, \log n\}}. \tag{A.2}$$

*Proof.* We split the matrix $A$ into two matrices, $B_1$ and $B_2$. $B_1$ consists of the block-diagonal projection onto the clusters, and $B_2$ is the rest. Denote the blocks on the diagonal of $B_1$ by $C_1$,

$C_2, \ldots, C_r$, where $C_i$ corresponds to the $i$th cluster. Then $\|B_1 - \mathbb{E}(B_1)\| = \max_i \|C_i - \mathbb{E}(C_i)\|$, and for each $i$, $\|C_i - \mathbb{E}(C_i)\| \gtrsim \sqrt{p_i(1-p_i)n_i}$ with probability at most $n_i^{-\alpha}$, by Lemma 5. By assumptions (1) and (2) of Lemma 7 and applying a union bound, we conclude that

$$\|B_1 - \mathbb{E}(B_1)\| \lesssim \max_i \sqrt{p_i(1-p_i)n_i}$$

with probability at least $1 - \sum_{i=1}^r n_i^{-\alpha} = 1 - o(1)$. We shall now turn our attention to $B_2$. Let $\sigma^2 = \max\{q(1-q), \log n/n\}$. By Corollary 6, $\|B_2 - \mathbb{E}(B_2)\| \lesssim \max\{\sqrt{q(1-q)n}, \sqrt{\log n}\}$, with high probability. Putting the two norm estimates together, the conclusion of Lemma 7 follows. ∎

We are now in the position to prove Theorem 1.

*Proof.* [of Theorem 1] We need to show that for any feasible $Y \neq Y^\star$, we have $\Delta(Y) := \langle A, Y^\star - Y \rangle > 0$. Rewrite $\Delta(Y)$ as

$$\Delta(Y) = \langle A, Y^\star - Y \rangle = \langle \mathbb{E}[A], Y^\star - Y \rangle + \langle A - \mathbb{E}[A], Y^\star - Y \rangle.$$

Note that $\sum_{i,j} Y_{ij}^\star = \sum_{i,j} Y_{ij} = \sum_{k=1}^r n_k^2$, thus $\sum_{i,j}(Y_{ij}^\star - Y_{ij}) = 0$. Express this as

$$\sum_{k=1}^r \sum_{i,j \in V_k} (Y^\star - Y)_{ij} = - \sum_{k' \neq k''} \sum_{i \in V_{k'}, j \in V_{k''}} (Y^\star - Y)_{ij}.$$

Then we have

$$\langle \mathbb{E}[A], Y^\star - Y \rangle = \sum_{k=1}^r \sum_{i,j \in V_k^\star} p_k (Y^\star - Y)_{ij} + \sum_{k' \neq k''} \sum_{i \in V_{k'}, j \in V_{k''}} q(Y^\star - Y)_{ij}$$

$$= \sum_{k=1}^r \sum_{i,j \in V_k} (p_k - q)(Y^\star - Y)_{ij}.$$

Finally, since $0 \leq (Y^\star - Y)_{ij} \leq 1$ for $i, j \in V_k$, we can write

$$\langle \mathbb{E}[A], Y^\star - Y \rangle = \sum_{k=1}^r \sum_{i,j \in V_k} (p_k - q)\|(Y^\star - Y)_{\mathcal{C}_k}\|_1. \tag{A.3}$$

Next, recall that the subdifferential (i.e., the set of all subgradients) of $\|\cdot\|_\star$ at $Y^\star$ is given by

$$\partial \|Y^\star\|_\star = \{UU^T + Z \mid U^T Z = ZU = 0, \ \|Z\| \leq 1\}$$

where $Y^\star = UKU^T$ is the singular value decomposition for $Y^\star$ with $U \in \mathbb{R}^{n \times r}$, $K = \text{diag}(n_1, \ldots, n_r)$, and $U_{ik} = 1/\sqrt{n_k}$ if node $i$ is in cluster $\mathcal{C}_k$ and $U_{ik} = 0$ otherwise.

Let $M := A - \mathbb{E}[A]$. Since conditions (1) and (2) of Lemma 7 are verified, there exists $C_1 > 0$ such that $\|M\| \leq \lambda$, with probability $1 - o(1)$, where

$$\lambda := C_1 \left( \max_i \sqrt{p_i(1-p_i)n_i} + \sqrt{\max\{q(1-q)n, \log n\}} \right). \tag{A.4}$$

Furthermore, let the projection operator onto a subspace $T$ be defined by

$$\mathcal{P}_T(M) := UU^T M + MUU^T - UU^T MUU^T,$$

and also $\mathcal{P}_{T^\perp} = \mathcal{I} - \mathcal{P}_T$, where $\mathcal{I}$ is the identity map. Since $\|\mathcal{P}_{T^\perp}(M)\| \leq \|M\| \leq \lambda$ with high probability, $UU^T + \frac{1}{\lambda}\mathcal{P}_{T^\perp}(M) \in \partial \|Y^\star\|_\star$ with high probability. Now, by the constraints of the convex program, we have

$$0 \geq \|Y\|_\star - \|Y^\star\|_\star$$
$$\geq \langle UU^T + \tfrac{1}{\lambda}\mathcal{P}_{T^\perp}(M), Y - Y^\star \rangle \tag{A.5}$$
$$= \langle UU^T - \tfrac{1}{\lambda}\mathcal{P}_T(M), Y - Y^\star \rangle + \tfrac{1}{\lambda}\langle M, Y - Y^\star \rangle,$$

which implies $\langle M, Y^\star - Y \rangle \geq \langle \mathcal{P}_T(M) - \lambda UU^T, Y^\star - Y \rangle$. Combining (A.2) and (A.3) we get,

$$
\begin{aligned}
\Delta(Y) &\geq \sum_{k=1}^{r}(p_k - q)\|(Y^\star - Y)_{\mathcal{C}_k}\|_1 + \langle \mathcal{P}_T(M) - \lambda UU^T, Y^\star - Y \rangle \\
&\geq \sum_{k=1}^{r}(p_k - q)\|(Y^\star - Y)_{\mathcal{C}_k}\|_1 \\
&\quad - \sum_{k=1}^{r}\underbrace{\|(\mathcal{P}_T(M) - \lambda UU^T)_{\mathcal{C}_k}\|_\infty}_{(\mu_{kk})}\|(Y^\star - Y)_{\mathcal{C}_k}\|_1 \\
&\quad - \sum_{k'\neq k''}\underbrace{\|(\mathcal{P}_T(M) - \lambda UU^T)_{V_{k'}\times V_{k''}}\|_\infty}_{(\mu_{k'k''})}\|(Y^\star - Y)_{V_{k'}\times V_{k''}}\|_1
\end{aligned}
\tag{A.6}
$$

where we have made use of the fact that an inner product can be bounded by a product of dual norms. We now derive bounds for the quantities $\mu_{kk}$ and $\mu_{k'k''}$ marked above. Note that the former indicates sums over the clusters, while the latter indicates sums outside the clusters.

For $\mu_{kk}$, if $(i,j) \in \mathcal{C}_k$ then

$$
\begin{aligned}
\left(\mathcal{P}_T(M) - \lambda UU^T\right)_{ij} &= \left(UU^T M + MUU^T - UU^T MUU^T - \lambda UU^T\right)_{ij} \\
&= \frac{1}{n_k}\sum_{l\in\mathcal{C}_k}M_{lj} + \frac{1}{n_k}\sum_{l\in\mathcal{C}_k}M_{il} - \frac{1}{n_k^2}\sum_{l,l'\in\mathcal{C}_k}M_{ll'} - \frac{\lambda}{n_k}.
\end{aligned}
$$

Recall Bernstein's inequality (e.g. see Theorem 1.6.1 in [37]):

**Proposition 8** *(Bernstein Inequality) Let $S_1, S_2, \ldots, S_n$ be independent, centered, real random variables, and assume that each one is uniformly bounded:*

$$
\mathbb{E}[S_k] = 0 \;\; and \;\; |S_k| \leq L \;\; for \; each \; k = 1,\ldots,n.
$$

*Introduce the sum $Z = \sum_{k=1}^{n}S_k$, and let $\nu(Z)$ denote the variance of the sum:*

$$
\nu(Z) = \mathbb{E}[Z^2] = \sum_{k=1}^{n}\mathbb{E}[S_k^2].
$$

*Then*

$$
\mathbb{P}[\,|Z| \geq t\,] \;\leq\; 2\exp\left(\frac{-t^2/2}{\nu(Z) + Lt/3}\right) \;\; for \; all \; t \geq 0.
$$

We will apply it to bound the three sums in $\mu_{kk}$, using the fact that each of the sums contains only centered, independent, and bounded variables, and that the variance of each entry in the sum is $p_k(1-p_k)$. For the first two sums, we can use $t \sim \sqrt{n_k p_k(1-p_k)\log n_k}$ to obtain a combined failure probability (over the entire cluster) of $O(n_k^{-\alpha})$. Finally, for the third sum, we may choose $t \sim n_k\sqrt{p_k(1-p_k)\log n_k}$, again for a combined failure probability over the whole cluster of no more than $O(n_k^{-\alpha})$.

We have thusly

$$
\begin{aligned}
\mu_{kk} &\leq |\tfrac{1}{n_k}\sum_{l\in\mathcal{C}_k}M_{lj}| + |\tfrac{1}{n_k}\sum_{l\in\mathcal{C}_k}M_{il}| + |\tfrac{1}{n_k^2}\sum_{l,l'}M_{l,l'}| + \frac{\lambda}{n_k} \\
&\lesssim \sqrt{\frac{p_k(1-p_k)}{n_k}}\log n_k + \frac{\sqrt{p_k(1-p_k)\log n_k}}{n_k} + \frac{\lambda}{n_k},
\end{aligned}
$$

for all $i,j \in \mathcal{C}_k$, with probability $1 - O(n_k^{-\alpha})$. Note that in the inequality above, the second term is much smaller in magnitude than the first, so we can disregard it; using (A.4), we obtain

$$
\mu_{kk} \lesssim \frac{1}{n_k}\left(\sqrt{n_k p_k(1-p_k)\log n_k} + \max_i\sqrt{p_i(1-p_i)n_i} + \sqrt{\max\{q(1-q)n, \log n\}}\right) \tag{A.7}
$$

and by taking a union bound over $k$ we can conclude that the probability that any of these bounds fail is $o(1)$. Similarly, for $\mu_{k'k''}$, for $k' \neq k''$, we can calculate that

$$\mu_{k'k''} \leq |\frac{1}{n_{k'}} \sum_{l \in \mathcal{C}_{k'}} M_{lj}| + |\frac{1}{n_{k''}} \sum_{l \in \mathcal{C}_{k''}} M_{il}| + |\frac{1}{n_{k'}n_{k''}} \sum_{l' \in \mathcal{C}_{k'}, l'' \in \mathcal{C}_{k''}} M_{l',l''}| \qquad \text{(A.8)}$$

$$\lesssim \sqrt{q(1-q)(\frac{\log n_{k'}}{n_{k'}} + \frac{\log n_{k''}}{n_{k''}})} + \frac{\sqrt{q(1-q)\log(n_{k'}n_{k''})}}{\sqrt{n_{k'}n_{k''}}},$$

with failure probability over all $i \in \mathcal{C}_{k'}$, $j \in \mathcal{C}_{k''}$ of no more than $O(n_{k'}^{-\alpha} n_{k''}^{-\alpha})$. We do this by taking $t \sim \sqrt{n_{k'}q(1-q)\log(n_{k'}n_{k''})}$, respectively $t \sim \sqrt{n_{k''}q(1-q)\log(n_{k'}n_{k''})}$ in the first two sums. For the third, we just take $t \sim \sqrt{n_{k'}n_{k''}q(1-q)\log(n_{k'}n_{k''})}$. As before, note that the second term is much smaller in magnitude than the first, and hence we can disregard it to obtain

$$\mu_{k'k''} \lesssim \max_k \sqrt{\frac{q(1-q)\log n_k}{n_k}} = \sqrt{\frac{q(1-q)\log n_{\min}}{n_{\min}}} := \mu_{\text{off}}, \qquad \text{(A.9)}$$

as the function $\log x / x$ is strictly increasing if $x \geq 3$, with the probability that all of the above are simultaneously true being $1 - o(1)$. Since the bound on $\mu_{k'k''}$ is independent of $k'$ and $k''$ we can rewrite (A.6) as

$$\Delta(Y) \geq \sum_{k=1}^{r}(p_k - q)\|(Y^\star - Y)_{\mathcal{C}_k}\|_1 - \sum_{k=1}^{r}\mu_{kk}\|(Y^\star - Y)_{\mathcal{C}_k}\|_1 - \sum_{k' \neq k''}\mu_{k'k''}\|(Y^\star - Y)_{V_{k'} \times V_{k''}}\|_1$$

$$\geq \sum_{k=1}^{r}(p_k - q - \mu_{kk} - \mu_{\text{off}})\|(Y^\star - Y)_{\mathcal{C}_k}\|_1$$

where we use the fact that $\sum_{k' \neq k''}\|(Y^\star - Y)_{V_{k'} \times V_{k''}}\|_1 = \sum_{k=1}^{r}\|(Y^\star - Y)_{\mathcal{C}_k}\|_1$. Finally, the conditions of theorem guarantee the nonnegativity of the right hand side, hence the optimality of $Y^\star$ as the solution to the convex recovery program in (2.4). ∎

### A.3 Proof of Theorem 2

We use a different result than Lemma 7, which we state below.

**Lemma 9 (Corollary 3.12 in [7])** *Let $X$ be an $n \times n$ symmetric matrix whose entries $X_{ij}$ are independent symmetric random variables. Then there exists for any $0 < \epsilon \leq \frac{1}{2}$ a universal constant $c_\epsilon$ such that for every $t \geq 0$*

$$\|X\| \leq 2(1+\epsilon)\tilde{\sigma} + t,$$

*with probability at least $1 - n\exp(\frac{-t^2}{c_\epsilon \tilde{\sigma}_\star^2})$, where*

$$\tilde{\sigma} = \max_i \sqrt{\sum_j \mathbb{E}[X_{ij}^2]}, \quad \tilde{\sigma}_\star = \max_{i,j}\|X_{ij}\|_\infty.$$

We specialize Lemma 9 to HSBM to get the following result.

**Lemma 10** *Let $A$ be generated according to the heterogenous stochastic block model (HSBM). Then there exists for any $0 < \epsilon \leq \frac{1}{2}$ a universal constant $c_\epsilon$ such that*

$$\|A - \mathbb{E}(A)\| \leq 4(1+\epsilon)\max\{\sigma_{\max}, \sigma_0\} + \sqrt{2c_\epsilon \log n} \qquad \text{(A.10)}$$

*with probability at least $1 - n^{-1}$.*

We can now present the proof for Theorem 2.

*Proof.* The proof follows the same lines as the proof of Theorem 1. Given the similarities between the proofs, we will only describe here the differences between the tools employed, and how they

affect the conditions in Theorem 2. The proof proceeds identically as before, up to the definition of $\lambda$, which–since we use Lemma 10 rather than 7–becomes

$$\lambda := C_2 \max\{\sigma_{\max}, \sigma_0, \sqrt{\log n}\}, \tag{A.11}$$

where $C_2$ was chosen as a good upper bounding constant for Lemma 10.

The other two small changes come from the fact that we will need to make sure that the failure probabilities for the quantities $\mu_{kk}$ and $\mu_{k'k''}$ are polynomial in $1/n$, which leads to the replacement of $\log n_k$ in either of them by a $\log n$. The rest of the proof proceeds exactly in the same way. ∎

## B    Proofs for Recoverability and Non-recoverability

We use the same notation as in the main paper and in Appendix A.1.

### B.1    Proofs for Recoverability

*Proof.* [of Theorem 3] For $\Delta(Y) := \langle A, Y^\star - Y \rangle$, we have to show that for any feasible $Y \neq Y^\star$, we have $\Delta(Y) > 0$. For simplicity we assume $Y_{ii} = Y_{ii}^\star = 0$ for all $i \in \{1, \ldots, n\}$. Consider an splitting as

$$\Delta(Y) = \langle A, Y^\star - Y \rangle = \langle \mathbb{E}(A), Y^\star - Y \rangle + \langle A - \mathbb{E}(A), Y^\star - Y \rangle. \tag{B.1}$$

Notice that $Y^\star = \sum_{k=1}^r \mathbf{1}_{\mathcal{C}_k}$ and $\mathbb{E}(A) = q\mathbf{1}\mathbf{1}^T + \sum_{k=1}^r (p_k - q)\mathbf{1}_{\mathcal{C}_k}$. Considering $d_k(Y) = \langle Y_{\mathcal{C}_k}^\star, Y^\star - Y \rangle$, the number of entries on $\mathcal{C}_k$ on which $Y$ and $Y^\star$ do not match, we get

$$\langle \mathbb{E}(A), Y^\star - Y \rangle = \sum_{k=1}^r (p_k - q)d_k(Y) \tag{B.2}$$

where we used the fact that $Y, Y^\star \in \mathcal{Y}$ and have the same number of ones and zeros, hence $\sum_{i,j} Y_{ij} = \sum_{i,j} Y_{ij}^\star$. On the other hand, the second term in (B.1) can be represented as

$$T(Y) := \langle A - \mathbb{E}(A), Y^\star - Y \rangle = \sum_{Y_{ij}^\star = 1, Y_{ij} = 0} (A - \mathbb{E}(A))_{ij} + \sum_{Y_{ij}^\star = 0, Y_{ij} = 1} (\mathbb{E}(A) - A)_{ij}$$

where each term is a centered Bernoulli random variable bounded by $1$. Observe that the total variance for all the summands in the above is given by

$$\sigma^2 = \sum_{k=1}^r d_k(Y)p_k(1 - p_k) + q(1 - q)\sum_{k=1}^r d_k(Y).$$

Then, combining (B.1) and (B.2), and applying the Bernstein inequality yields

$$\mathbb{P}(\Delta(Y) \leq 0) = \mathbb{P}\left(T(Y) \leq -\sum_k (p_k - q)d_k(Y)\right) \leq \exp\left(-\frac{t^2}{2\sigma^2 + 2t/3}\right) = \exp\left(-\frac{\sum_k (p_k - q)d_k(Y)}{2\nu(Y) + 2/3}\right)$$

where $t = \sum_k (p_k - q)d_k(Y)$ and

$$\begin{aligned}
\nu(Y) &= \frac{\sigma^2}{t} \\
&= \frac{\sum_{k=1}^r (p_k(1 - p_k) + q(1 - q))d_k(Y)}{\sum_k (p_k - q)d_k(Y)} \\
&\leq \max_k \frac{p_k(1 - p_k) + q(1 - q)}{p_k - q} \\
&= \frac{p_{\min}(1 - p_{\min}) + q(1 - q)}{p_{\min} - q} := \bar{\nu}_0.
\end{aligned}$$

Considering $\bar{\nu} := 2\bar{\nu}_0 + 2/3$ and $\theta_k := \lfloor \frac{p_k - q}{p_{\min} - q} \rfloor$, we get

$$\mathbb{P}(\Delta(Y) \leq 0) \leq \exp\left(-\frac{1}{\bar{\nu}}\sum_k (p_k - q)d_k(Y)\right) \leq \exp\left(-\frac{1}{\bar{\nu}}(p_{\min} - q)\sum_k \theta_k d_k(Y)\right) \tag{B.3}$$

which can be bounded using the next lemma which is a direct extension of Lemma 4 in [15].

**Lemma 11** *Given the values of $\theta_k$ and $n_k$, for $k = 1, \ldots, r$, and for each integer value $\xi \in [\min \theta_k(2n_k - 1), \sum_k \theta_k n_k^2]$, we have*

$$\left|\{[Y] \subset \mathcal{Y} : \sum_{k=1}^{r} \theta_k d_k(Y) = \xi\}\right| \leq \left(\frac{5\xi}{\tau}\right)^2 n^{16\xi/\tau} \tag{B.4}$$

*where $\tau := \min_k \theta_k n_k$, and $[Y] = \{Y' \in \mathcal{Y} : Y'_{ij} Y^\star_{ij} = Y_{ij} Y^\star_{ij}\}$.*

Now plugging in the result of Lemma 11 into (B.3) yields,

$$\mathbb{P}\left(\exists Y \in \mathcal{Y} : Y \neq Y^\star, \Delta(Y) \leq 0\right) \leq \sum_\xi \mathbb{P}\left(\exists Y \in \mathcal{Y} : \sum_k \theta_k d_k(Y) = \xi, \Delta(Y) \leq 0\right)$$

$$\leq 2 \sum_\xi \left(\frac{5\xi}{\tau}\right)^2 n^{16\xi/\tau} \exp\left(-\tfrac{1}{\nu}(p_{\min} - q)\xi\right)$$

$$= 50 \sum_\xi \left(\frac{\xi}{\tau}\right)^2 \exp\left((16 \log n - \tfrac{1}{\nu}(p_{\min} - q)\tau)\frac{\xi}{\tau}\right)$$

$$\leq 50 \sum_\xi \left(\frac{\xi}{\tau}\right)^2 \exp\left((16 \log n - \tfrac{1}{2\nu}\rho_{\min})\frac{\xi}{\tau}\right) \tag{B.5}$$

In order to have a meaningful bound for the above probability, we need the exponential term in (B.5) to be decreasing. Hence, we require $\rho_{\min} \geq 64\bar{\nu} \log n$. Moreover, the function in (B.5) is a decreasing function of $\xi/\tau$ for

$$\frac{\xi}{\tau} \geq \frac{4\bar{\nu}}{\rho_{\min} - 32\bar{\nu} \log n}. \tag{B.6}$$

Since $\xi \geq \min \theta_k(2n_k - 1) \geq \min \theta_k n_k = \tau$, requiring the following condition (for some $\eta > 0$ which will be determined later),

$$\rho_{\min} \geq 2(16 + \eta)\bar{\nu} \log n + 4\bar{\nu}, \tag{B.7}$$

implies

$$\frac{\xi}{\tau} \geq 1 \geq \frac{4}{4 + 2\eta \log n} \geq \frac{4\bar{\nu}}{\rho_{\min} - 32\bar{\nu} \log n}$$

and allows us to bound the summation in (B.5) with the largest term (corresponding to the smallest value of $\xi/\tau$, or an even smaller value, namely 1) times the number of summands (which is bounded by $\sum \theta_k n_k^2$ since $\theta_k$'s are integers); i.e.,

$$(\text{B.5}) \leq 50 \left(\sum \theta_k n_k^2\right) \exp\left(16 \log n - \tfrac{1}{2\nu}\rho_{\min}\right) \tag{B.8}$$

$$\leq 50 \sum \theta_k n_k^2 \exp(-2 - \eta \log n) \tag{B.9}$$

$$\leq 7\, \theta_{\max} n^{2-\eta} \tag{B.10}$$

$$\leq 7\, \frac{p_{\max} - q}{p_{\min} - q} n^{2-\eta}, \tag{B.11}$$

or, similarly,

$$(\text{B.5}) \leq 50 \sum \theta_k n_k^2 \exp(-2 - \eta \log n) \leq 7 \frac{\sum_{k=1}^{r} \rho_k}{p_{\min} - q} n^{1-\eta}. \tag{B.12}$$

Hence, if the condition in (B.7) holds we get the optimality of $Y^\star$ with a probability at least equal to the above. Finally, $n \geq 8$ implies $\log n \geq 2$ and (B.7) follows from

$$\rho_{\min} \geq 4(17 + \eta)\left(\frac{1}{3} + \frac{p_{\min}(1 - p_{\min}) + q(1 - q)}{p_{\min} - q}\right) \log n.$$

∎

*Proof.* [of Lemma 11] We extend the proof of Lemma 4 in [15] to our case. Fix a $Y \in \mathcal{Y}$ with $\sum_{k=1}^{r} \theta_k d_k(Y) = \xi$ and consider the corresponding $r$ clusters as well as the set of isolated nodes. Notice that for any $Y' \in [Y]$ we also have $\sum_{k=1}^{r} \theta_k d_k(Y') = \xi$. In the following, we will construct an ordering for the clusters of $Y$ based on $Y^\star$. Denote the clusters of $Y^\star$ by $V_1^\star, \ldots, V_r^\star$, and $V_{r+1}^\star$.

Consider the set of values of cluster sizes $\{n_1, \ldots, n_r\} = \{\eta_1, \ldots, \eta_s\}$ where $\eta_1, \ldots, \eta_s$ are distinct, and define $\mathcal{I}_\ell = \{k : n_k = \eta_\ell\} \subset \{1, \ldots, r\}$ for $\ell = 1, \ldots, s$. For any $\ell$ with $|\mathcal{I}_\ell| = 1$, the cluster in $Y \in \mathcal{Y}$ of size $\eta_\ell$ can be uniquely assigned to a cluster among $V_1^\star, \ldots, V_r^\star$ of similar size. We now define an ordering for the remaining clusters. Consider a $\ell$ with $|\mathcal{I}_\ell| > 1$, and restrict the attention to clusters $V$ of size $\eta_\ell$ and clusters $V_k^\star$ for $k \in \mathcal{I}$ (all clusters in $Y^\star$ of size $\eta_\ell$). This is similar to the case in [15] where all sizes are equal: For each new cluster $V$ of size $\eta_\ell$, if there exists a $k \in \mathcal{I}_\ell$ such that $|V \cap V_k^\star| > \frac{1}{2}\eta_\ell$ then we label this cluster as $V_k$; this label is unique. The remaining unlabeled clusters are labeled arbitrarily by a number in $\mathcal{I}_\ell$.

Hence, we labeled all the clusters of $Y$ according to the clusters of $Y^\star$. For each $(k, k') \in \{1, \ldots, r\} \times \{1, \ldots, r+1\}$, we use $\alpha_{kk'} := |V_k^\star \cap V_{k'}|$ to denote the sizes of intersections of clusters of $Y$ and $Y^\star$. We observe that the new clusters $(V_1, \ldots, V_{r+1})$ have the following properties:

(A1) $(V_1, \ldots, V_{r+1})$ is a partition of $\{1, \ldots, n\}$ with $|V_k| = n_k$ for all $k = 1, \ldots, r$; since $Y \in \mathcal{Y}$.

(A2) For $\ell \in \{1, \ldots, s\}$ with $|\mathcal{I}_\ell| = 1$, we have $\alpha_{kk} = n_k$ for the index $k \in \mathcal{I}_\ell$.

(A3) For $\ell \in \{1, \ldots, s\}$ with $|\mathcal{I}_\ell| > 1$, consider any $k \in \mathcal{I}_\ell$. Then, exactly one of the following is true: (1) $\alpha_{kk} > \frac{1}{2}n_k$; (2) $\alpha_{kk'} \leq \frac{1}{2}n_k$ for all $k' \in \mathcal{I}_\ell$.

(A4) For $d_k(Y) = \langle Y_{\mathcal{C}_k}^\star, Y^\star - Y \rangle$, where $k = 1 \ldots, r$, we have

$$d_k(Y) = |\{(i, j) : (i, j) \in \mathcal{C}_k^\star, Y_{ij} = 0\}|$$
$$= |\{(i, j) : (i, j) \in \mathcal{C}_k^\star, i, j \in V_{r+1}\}|$$
$$+ \sum_{k' \neq k''} |\{(i, j) : (i, j) \in \mathcal{C}_k^\star, (i, j) \in V_{k'} \times V_{k''}\}|$$
$$= \alpha_{k(r+1)}^2 + \sum_{k' \neq k''} \alpha_{kk'} \alpha_{kk''},$$

which implies

$$\xi = \sum_{k=1}^{r} \theta_k d_k(Y) = \sum_{k=1}^{r} \theta_k \alpha_{k(r+1)}^2 + \sum_{k=1}^{r} \sum_{k' \neq k''} \theta_k \alpha_{kk'} \alpha_{kk''}.$$

Unless specified otherwise, all the summations involving $k'$ or $k''$ are over the range $1, \ldots, r+1$.

We showed that the ordered partition for a $Y \in \mathcal{Y}$ with $\sum_{k=1}^{r} \theta_k d_k(Y) = \xi$ satisfies the above properties. Therefore,

$$|\{[Y] \in \mathcal{Y} : \sum_{k=1}^{r} \theta_k d_k(Y) = \xi\}| \leq |\{(V_1, \ldots, V_{r+1}) \text{ satisfying the above conditions}\}|.$$

Next, we upper bound the right hand side of the above.

Fix an ordered clustering $(V_1, \ldots, V_{r+1})$ which satisfies the above conditions. Define,

$$m_1 := \sum_{k' \neq 1} \alpha_{1k'}$$

as the number of nodes in $V_1^\star$ that are misclassified by $Y$; hence $m_1 + \alpha_{11} = n_1$. Consider the following two cases:

- if $\alpha_{11} > n_1/4$ we have

$$\sum_{k' \neq k''} \alpha_{1k'} \alpha_{1k''} \geq \alpha_{11} \sum_{k'' \neq 1} \alpha_{1k''} > \tfrac{1}{4} n_1 m_1$$

- if $\alpha_{11} \leq n_1/4$ we have $m_1 \geq 3n_1/4$, which from the aforementioned properties, we must have $\alpha_{1k'} \leq n_1/2$ for all $k' = 1, \ldots, r$. Then,

$$\sum_{k' \neq k''} \alpha_{1k'}\alpha_{1k''} + \alpha_{1(r+1)}^2 \geq \sum_{1 \neq k' \neq k'' \neq 1} \alpha_{1k'}\alpha_{1k''} + \alpha_{1(r+1)}^2 = m_1^2 - \sum_{k'=2}^{r} \alpha_{1k'}^2 \geq m_1^2 - \tfrac{1}{2}n_1 m_1 \geq \tfrac{1}{4}n_1 m_1$$

Therefore,

$$d_1(Y) = \sum_{k' \neq k''} \alpha_{1k'}\alpha_{1k''} + \alpha_{1(r+1)}^2 \geq \tfrac{1}{4}n_1 m_1$$

which holds for all other indices $k \neq 1$ as well. This yields

$$\xi \geq \tfrac{1}{4}\sum_{k=1}^{r} \theta_k n_k m_k \geq \tfrac{1}{4}(\min_k \theta_k n_k)\sum_{k=1}^{r} m_k \implies \bar{w} := \sum_{k=1}^{r} m_k \leq \frac{4\xi}{\min_k \theta_k n_k} := M$$

where $\bar{w}$ is the number of misclassified non-isolated nodes. Since one misclassified isolated node produces one misclassified non-isolated node, we have $w_0 \leq \bar{w} \leq M$ where $w_0$ is the number of misclassified isolated nodes.

- The pair of numbers $(\bar{w}, w_0)$ can take at most $(M+1)^2$ different values.
- For each such pair of numbers, there are at most $\bar{n}^{2M}$ ways to choose the identity of the misclassified nodes.
- Each misclassified non-isolated node can be assigned to one of $r - 1 \leq \bar{n}$ different clusters or be left isolated, and each misclassified isolated node can be assigned to one of $r \leq \bar{n}$ clusters.

All in all,

$$\left|\{[Y] \in \mathcal{Y} : \sum_{k=1}^{r} \theta_k d_k(Y) = \xi\}\right| \leq (M+1)^2 \bar{n}^{4M}$$

$$= \left(\frac{4\xi}{\min_k \theta_k n_k} + 1\right)^2 \exp\left(\frac{16\xi}{\min_k \theta_k n_k} \log \bar{n}\right)$$

$$\leq \left(\frac{5\xi}{\min_k \theta_k n_k}\right)^2 \exp\left(\frac{16\xi}{\min_k \theta_k n_k} \log \bar{n}\right).$$

$\blacksquare$

### B.2 Proofs for impossibility of recovery

We prove a more comprehensive version of Theorem 4.

**Theorem 12** *If any of the following conditions holds,*

*(1)* $2 \leq n_k \leq n/e$, *and*

$$4\sum_{k=1}^{r} n_k^2 \widetilde{D}(p_k, q) \leq \tfrac{1}{2}\sum_k n_k \log \tfrac{n}{n_k} - r - 2 \tag{B.13}$$

*(2)* $2 \leq n_k \leq n/e$, *and*

$$\tfrac{1}{2}r + \log \tfrac{1-p_{\min}}{1-p_{\max}} + 1 + \sum_k n_k^2 p_k \leq (\tfrac{1}{4}n - \sum n_k^2 p_k)\log n + \sum (n_k p_k - \tfrac{1}{4})n_k \log n_k \tag{B.14}$$

*(3)* $n \geq 128$, $r \geq 2$ *and*

$$\max_k n_k \left(\widetilde{D}(p_k, q) + \widetilde{D}(q, p_k)\right) \leq \tfrac{1}{12}\log(n - n_{\min}) \tag{B.15}$$

*then*

$$\inf_{\hat{Y}} \sup_{Y^\star \in \mathcal{Y}} \mathbb{P}[\hat{Y} \neq Y^\star] \geq \frac{1}{2}$$

*where the infimum is taken over all measurable estimators $\hat{Y}$ based on the realization $A$ generated according to the heterogenous stochastic block model.*

*Proof.* [of cases 1 and 2 of Theorem 12] Let $\mathbb{P}_{(Y^\star, A)}$ be the joint distribution of $Y^\star$ and $A$, where $Y^\star$ is sampled uniformly from $\mathcal{Y}$ and $A$ is generated according to the heterogenous stochastic block model conditioning on $Y^\star$. Note that

$$\inf_{\hat{Y}} \sup_{Y^\star \in \mathcal{Y}} \mathbb{P}[\hat{Y} \neq Y^\star] \geq \inf_{\hat{Y}} \mathbb{P}_{(Y^\star, A)}[\hat{Y} \neq Y^\star].$$

By Fano's inequality we have,

$$\mathbb{P}_{(Y^\star, A)}[\hat{Y} \neq Y^\star] \geq 1 - \frac{I(Y^\star; A) + 1}{\log |\mathcal{Y}|}, \tag{B.16}$$

where $I(X; Z)$ is the mutual information, and $H(X)$ is the Shannon entropy for $X$. By counting argument we find that $|\mathcal{Y}| = \binom{n}{\bar{n}} \frac{\bar{n}!}{n_1! \ldots n_r!}$. Using $\sqrt{n}(n/e)^n \leq n! \leq e\sqrt{n}(n/e)^n$ and $\binom{n}{\bar{n}} \geq (n/\bar{n})^{\bar{n}}$, it follows that

$$|\mathcal{Y}| \geq \frac{n^{\bar{n}} \sqrt{\bar{n}}}{e^r \sqrt{n_1 \ldots n_r} n_1^{n_1} \ldots n_r^{n_r}}$$

which gives

$$\log |\mathcal{Y}| \geq \sum_{i=1}^{r} n_i \Big( \log \frac{n}{n_i} - \frac{\log n_i}{2 n_i} \Big) - r \geq \frac{1}{2} \sum_{i=1}^{r} n_i \log \frac{n}{n_i} - r.$$

On the other hand, note that $H(A) \leq \binom{n}{2} H(A_{12})$ by chain rule, the fact that $H(X|Y) \leq H(X)$, and the symmetry among identically distributed $A_{ij}$'s. Furthermore $A_{ij}$'s are conditionally independent and hence $H(A|Y^\star) = \binom{n}{2} H(A_{12}|Y_{12}^\star)$. Now it follows that

$$I(Y^\star; A) = H(A) - H(A|Y^\star) \leq \binom{n}{2} I(Y_{12}^\star; A_{12}).$$

Observe that

$$\mathbb{P}(Y_{12}^\star = 1, (1,2) \in \mathcal{C}_i) = \frac{\binom{n-2}{n_i-2} \binom{n-n_i}{n_1,\ldots,n_{i-1},n_{i+1},\ldots,n_r,n_0}}{|\mathcal{Y}|} = \frac{n_i(n_i - 1)}{n(n-1)} := \alpha_i.$$

Using the properties of KL-divergence, we have $\mathbb{P}(A_{12} = 1) = \sum_{i=1}^{r} \alpha_i p_i + (1 - \sum_i \alpha_i) q := \beta$. Therefore,

$$\begin{aligned} I(Y_{12}^\star, A_{12}) &= \sum_{i=1}^{r} \alpha_i D_{\mathrm{KL}}(p_i, \beta) + (1 - \sum_i \alpha_i) D_{\mathrm{KL}}(q, \beta) \\ &= H(\beta) - \sum \alpha_i H(p_i) - (1 - \sum \alpha_i) H(q) \end{aligned} \tag{B.17}$$

Since $I(Y^\star; A) \leq \binom{n}{2} I(Y_{12}^\star; A_{12})$, plugging in the following condition in Fano's inequality (B.16),

$$\Big( \frac{1}{2} \sum_i n_i \log \frac{n}{n_i} - r \Big) \geq 2 + 2 \binom{n}{2} I(Y_{12}^\star; A_{12}), \tag{B.18}$$

guarantees $\mathbb{P}_{(Y^\star, A)}(\hat{Y} \neq Y^\star) \geq \frac{1}{2}$. In the following, we bound $I(Y_{12}^\star; A_{12})$ in two different ways to derive conditions 1 and 2 of Theorem 12. Throughout the proof we use the following inequality from [15] for the Kullback-Leibler divergence of Bernoulli variables,

$$D_{\mathrm{KL}}(p, q) := D_{\mathrm{KL}}(\mathrm{Ber}(p), \mathrm{Ber}(q)) = p \log \frac{p}{q} + (1-p) \log \frac{1-p}{1-q} \leq \frac{(p-q)^2}{q(1-q)}, \tag{B.19}$$

where the inequality is established by $\log x \leq x - 1$, for any $x \geq 0$.

- From (B.17), we have

$$I(Y_{12}^\star, A_{12}) \le \sum_{i=1}^r \frac{4\alpha_i(p_i - q)^2}{q(1 - q)} \le \frac{4\sum_{i=1}^r n_i^2(p_i - q)^2}{n(n-1)q(1-q)} \tag{B.20}$$

where we assumed $\sum n_i^2 \le \frac{1}{2}n^2$. Now, the right hand side of B.18 can be bounded as

$$2\binom{n}{2}I(Y_{12}^\star; A_{12}) \le \frac{4\sum_{i=1}^r n_i^2(p_i - q)^2}{q(1 - q)} = 4\sum_{i=1}^r n_i^2 \widetilde{D}(p_i, q)$$

and gives the sufficient condition 1 of Theorem 12.

- Again from (B.17), we have

$$I(Y_{12}^\star; A_{12}) = \sum_i \alpha_i \left( p_i \log \frac{p_i}{\beta} + (1 - p_i) \log \frac{1 - p_i}{1 - \beta} \right) + \left( 1 - \sum_i \alpha_i \right) D_{\mathrm{KL}}(q, \beta)$$

$$\le \sum_i \alpha_i p_i \log \frac{1}{\alpha_i} + \log c + \left( 1 - \sum_i \alpha_i \right) \frac{(q - \beta)^2}{\beta(1 - \beta)}$$

where the first term is bounded via $\beta \ge \sum_i \alpha_i p_i \ge \alpha_i p_i$, the second term is bounded via $\beta \le p_{\max}$ and $c = (1 - p_{\min})/(1 - p_{\max})$, and we used (B.19) for the last term. Since $1 - \beta = 1 - q - \sum_i \alpha_i(p_i - q) \ge (1 - \sum_i \alpha_i)(1 - q)$, the last term can be bounded as

$$\left( 1 - \sum_i \alpha_i \right) \frac{(q - \beta)^2}{\beta(1 - \beta)} \le \left( 1 - \sum_i \alpha_i \right) \frac{\left( \sum_i \alpha_i(p_i - q) \right)^2}{\left( \sum_i \alpha_i p_i \right)\left( 1 - \sum_i \alpha_i \right)(1 - q)} \le \sum_i \alpha_i(p_i - q) \le \sum_i \alpha_i p_i.$$

This implies

$$I(Y_{12}^\star; A_{12}) \le \sum_i \alpha_i p_i \log \frac{1}{\alpha_i} + \sum_i \alpha_i p_i + \log c \le \sum_i \alpha_i p_i \log \frac{e}{\alpha_i} + \log c. \tag{B.21}$$

Since $n_i \ge 2$, $\alpha_i = \frac{n_i(n_i-1)}{n(n-1)} \ge \frac{n_i^2}{en^2}$. Hence

$$2\binom{n}{2}I(Y_{12}^\star; A_{12}) \le n(n-1)\sum_i \frac{n_i(n_i-1)}{n(n-1)} p_i \log \frac{e^2 n^2}{n_i^2} + 2\log c \le 2\sum_i n_i^2 p_i \log \frac{en}{n_i} + 2\log c$$

which gives the sufficient condition 2 of Theorem 12.

∎

*Proof.* [of case 3 in Theorem 12] Without loss of generality assume $n_1 \le n_2 \le \ldots \le n_r$. Let $M := \bar{n} - n_{\min} = \bar{n} - n_1$, and $\bar{\mathcal{Y}} := \{Y_0, Y_1, \ldots, Y_M\}$. $Y_0$ is the clustering matrix with clusters $\{\mathcal{C}_\ell\}_{\ell=1}^r$ that correspond to $V_1 = \{1, \ldots, n_1\}$, $V_\ell = \{\sum_{i=1}^{\ell-1} n_i + 1, \ldots, \sum_{i=1}^\ell n_i\}$ for $\ell = 2, \ldots, r$. Other members of $\bar{\mathcal{Y}}$ are given by swapping an element of $\cup_{\ell=2}^r V_\ell$ with an element of $V_1$. Let $\mathbb{P}_i$ be the distributional law of the graph $A$ conditioned on $Y^\star = Y_i$. Since $\mathbb{P}_i$ is product of $\frac{1}{2}n(n-1)$ Bernoulli random variables, we have

$$I(Y^\star; A) = \mathbb{E}_Y \left[ D_{\mathrm{KL}}\left( \mathbb{P}(A|Y), \mathbb{P}(A) \right) \right]$$

$$= \frac{1}{M+1} \sum_{i=0}^M D_{\mathrm{KL}}\left( \mathbb{P}_i, \frac{1}{M+1} \sum_{j=0}^M \mathbb{P}_j \right)$$

$$\le \frac{1}{(M+1)^2} \sum_{i,j=0}^M D_{\mathrm{KL}}(\mathbb{P}_i, \mathbb{P}_j) \tag{B.22}$$

$$\le \max_{i,j=0,\ldots,M} D_{\mathrm{KL}}(\mathbb{P}_i, \mathbb{P}_j)$$

$$\le \max_{i_1,i_2,i_3=1,\ldots,r} \sum_{j=1}^3 \left( \frac{n_{i_j}(p_{i_j} - q)^2}{q(1 - q)} + \frac{n_{i_j}(p_{i_j} - q)^2}{p_{i_j}(1 - p_{i_j})} \right)$$

$$\le 3\max_{i=1,\ldots,r} \left( \frac{n_i(p_i - q)^2}{q(1 - q)} + \frac{n_i(p_i - q)^2}{p_i(1 - p_i)} \right)$$

where the third line follows from the convexity of KL-divergence, and the line before the last follows from the construction of $\bar{\mathcal{Y}}$ and (B.19). Now if the condition of the theorem holds, then $I(Y^\star; A) \leq \frac{1}{4}\log(n - n_{\min}) = \frac{1}{4}\log|\bar{\mathcal{Y}}|$. Note that for $n \geq 128$ we get $\log|\bar{\mathcal{Y}}| = \log(n - n_{\min}) \geq \log(n/2) \geq 4$. The conclusion follows by Fano's inequality in (B.16) restricting the supremum to be taken over $\bar{\mathcal{Y}}$. ∎

## C   Recovery by a Simple Counting Algorithm

In Section 2.1, we considered a tractable approach for exact recovery of (partially) observed models generated according to the heterogenous stochastic block model. However, in the interest of computational effort, one can further characterize a subset of models that are recoverable via a much simpler method than the convex program. The following algorithm is a proposal to do so. Moreover, the next theorem provides a characterization for models for which this simple thresholding algorithm is effective for exact recovery. Here, we allow for isolated nodes as described in Section 2.

---
**Algorithm 1** SIMPLE THRESHOLDING ALGORITHM
---
1: (Find isolated notes) For each node $v$, compute its degree $d_v$. Declare $i$ as isolated if

$$d_v < \min_k \frac{(n_k - 1)(p_k - q)}{2} + (n - 1)q.$$

2: (Find all communities) For every pair of nodes $(v, u)$, compute the number of common neighbors $S_{vu} := \sum_{w \neq v, u} A_{vw} A_{uw}$. Declare $v, u$ as in the same community if

$$S_{vu} > nq^2 + \frac{1}{2}\left(\min_k\left((n_k - 2)p_k^2 - n_k q^2\right) + q \cdot \max_{i \neq j}\left(\rho_k - p_k + \rho_l - p_l\right)\right)$$

where $\rho_k = n_k(p_k - q)$.

---

**Theorem 13** *Under the stochastic block model, with probability at least $1 - 2n^{-1}$, the simple counting algorithm 1 find the isolated nodes provided*

$$\min_k (n_k - 1)^2 (p_k - q)^2 \geq 19(1 - q)\left(\max_k n_k p_k + nq\right)\log n. \tag{C.1}$$

*Furthermore the algorithm finds the cluster if*

$$\left[\min_k\left\{(n_k - 2)p_k^2 + (n - n_k)q^2\right\} - q\max_{k \neq l}\left\{(n_k - 1)p_k + (n_l - 1)p_l + (n - n_k - n_l)q\right\}\right]^2$$
$$\geq 26(1 - q^2)\left(\max_k n_k p_k^2 + nq^2\right)\log n, \tag{C.2}$$

*while the term inside the bracket (which is squared) is assumed to be non-negative.*

We remark that the following is a slightly more restrictive condition than (C.2)

$$\left[\min_k n_k(p_k^2 - q^2) - 2q\rho_{\max}\right]^2 \geq 26(1 - q^2)\left[nq^2 + \max_k n_k p_k^2\right]\log n. \tag{C.3}$$

with better interpretability.

*Proof.* [of Theorem 13] For node $v$, let $d_v$ denote its degree. Let $\bar{V} = \cup_{i=1}^r V_i$ denote the set of nodes which belong to one of the clusters, and $V_0$ be isolated nodes. If $v \in V_i$ for some $i = 1, \ldots, r$, then $d_v$ is distributed as a sum of independent binomial random variables $\text{Bin}(n_i - 1, p_i)$ and $\text{Bin}(n - n_i, q)$. If $v \in V_0$, then $d_v$ is distributed as $\text{Bin}(n - 1, q)$. Hence we have,

$$\mathbb{E}[d_v] = \begin{cases} (n_i - 1)p_i + (n - n_i)q & v \in V_i \subset \bar{V} \\ (n - 1)q & v \in V_0, \end{cases}$$

and

$$\text{Var}[d_v] = \begin{cases} (n_i - 1)p_i(1 - p_i) + (n - n_i)q(1 - q) & v \in V_i \subset \bar{V} \\ (n - 1)q(1 - q) & v \in V_0 . \end{cases}$$

Let $\kappa_0^2 := \max_i n_i p_i(1 - q) + nq(1 - q)$, and $t = \min_i \frac{(n_i - 1)(p_i - q)}{2} \leq \frac{\kappa_0^2}{2}$. Then $\text{Var}[d_v] \leq \kappa_0^2$ for any $v \in V_0 \cup \bar{V}$. By Bernstein's inequality we get

$$\mathbb{P}\left[|d_v - \mathbb{E}[d_v]| > t\right] \leq 2\exp\left(-\frac{t^2}{2\kappa_0^2 + 2t/3}\right) \leq 2\exp\left(-\frac{3\min_i(n_i - 1)^2(p_i - q)^2}{28\kappa_0^2}\right) \leq 2n^{-2}, \tag{C.4}$$

where the last inequality follows from the condition (C.1). Now by union bound over all nodes, with probability at least $1 - 2n^{-1}$, for node $v \in V_i \subset \bar{V}$ we have,

$$d_v \geq (n_i - 1)p_i + (n - n_i)q - t > \min_i \frac{(n_i - 1)(p_i - q)}{2} + (n - 1)q, \tag{C.5}$$

and for node $v \in V_0$,

$$d_v \leq (n - 1)q(1 - q) + t < \min_i \frac{(n_i - 1)(p_i - q)}{2} + (n - 1)q. \tag{C.6}$$

This proves the first statement of the theorem, and all the isolated nodes are correctly identified. For the second statement, let $S_{vu}$ denote the common neighbor for nodes $v, u \in \bar{V}$. Then

$$S_{vu} \sim_d \begin{cases} \text{Bin}(n_i - 2, p_i^2) + \text{Bin}(n - n_i, q^2) & (v, u) \in V_i \times V_i \\ \text{Bin}(n_i - 1, p_i q) + \text{Bin}(n_j - 1, p_j q) + \text{Bin}(n - n_i - n_j, q^2) & (v, u) \in V_i \times V_j, \ i \neq j \end{cases}$$

where $\sim_d$ denotes equality in distribution and $+$ denotes the summation of independent random variables. Hence

$$\mathbb{E}[S_{vu}] = \begin{cases} (n_i - 2)p_i^2 + (n - n_i)q^2 & (v, u) \in V_i \times V_i \\ (n_i - 1)p_i q + (n_j - 1)p_j q + (n - n_i - n_j)q^2 & (v, u) \in V_i \times V_j, \ i \neq j \end{cases}$$

and

$$\text{Var}[S_{vu}] = \begin{cases} (n_i - 2)p_i^2(1 - p_i^2) + (n - n_i)q^2(1 - q^2) & (v, u) \in V_i \times V_i \\ (n_i - 1)p_i q(1 - p_i q) + (n_j - 1)p_j q(1 - p_j q) \\ \quad + (n - n_i - n_j)q^2(1 - q^2) & (v, u) \in V_i \times V_j, \ i \neq j \end{cases}$$

Let

$$\Delta = \min_i \left((n_i - 2)p_i^2 + (n - n_i)q^2\right) - \max_j \left(2(n_j - 1)p_j q + (n - 2n_j)q^2\right)$$
$$= \min_i \left((n_i - 2)p_i^2 - n_i q^2\right) - \max_j \left(2(n_j - 1)p_j q - 2n_j q^2\right),$$

Let $\kappa_1^2 := 2\max_i n_i p_i^2(1 - q^2) + nq^2(1 - q^2)$. Then $\text{Var}[S_{vu}] \leq \kappa_1^2$ for all $v, u$. Then $\Delta \leq \kappa_1^2/2$. Bernstein's inequality with $t = \Delta/2$ yields

$$\mathbb{P}\left[|S_{vu} - \mathbb{E}[S_{vu}]| > t\right] \leq 2\exp\left(-\frac{t^2}{2\kappa_1^2 + 2t/3}\right) \leq 2\exp\left(-\frac{3\Delta^2}{26\kappa_1^2}\right) \leq 2n^{-3}, \tag{C.7}$$

where the last line follows from assumption (C.2). By union bound over all pair of nodes $(v, u)$, we get with probability at least $1 - 2n^{-1}$, $S_{vu} > \Gamma$ for all $v, u$ in the same cluster and $S_{vu} < \Gamma$ otherwise. Here

$$\Gamma := \frac{1}{2}\left(\min_i \left((n_i - 2)p_i^2 + (n - n_i)q^2\right) + \max_{i \neq j} \left((n_i - 1)p_i q + (n_j - 1)p_j q + (n - n_i - n_j)q^2\right)\right).$$

■

# D Detailed Computations for Examples in Section 3

In the following, we present the detailed computations for the examples in Section 3 and summarized in Table 1. When there is no impact on the final result, quantities are approximated as denoted by $\approx$.

First, we repeat the conditions of Theorems 1 and 2. The conditions of Theorem 1 can be equivalently stated as

- $\rho_k^2 \gtrsim n_k p_k (1-p_k) \log n_k = \sigma_k^2 \log n_k$
- $(p_{\min} - q)^2 \gtrsim q(1-q) \frac{\log n_{\min}}{n_{\min}}$
- $\rho_{\min}^2 \gtrsim \max\{\log n, nq(1-q), \max_k n_k p_k (1-p_k)\}$
- $\sum_{k=1}^{r} n_k^{-\alpha} = o(1)$ for some $\alpha > 0$.

Notice that $n_k p_k (1-p_k) \gtrsim \log n_k$, for $k = 1, \ldots, r$, is implied by the first condition, as mentioned in Remark 1. The conditions of Theorem 2 can be equivalently stated as

- $\rho_k^2 \gtrsim n_k p_k (1-p_k) \log n$
- $(p_{\min} - q)^2 \gtrsim q(1-q) \frac{\log n}{n_{\min}}$
- $\rho_{\min}^2 \gtrsim \max\{nq(1-q), \max_k n_k p_k (1-p_k)\}$.

**Remark 2** *Provided that both $p_k$ and $q/p_k$ are bounded away from $1$, we have*

$$\widetilde{D}(q, p_k) = p_k \frac{(1-q/p_k)^2}{1-p_k} \approx p_k \quad , \quad \frac{\rho_k^2}{\sigma_k^2} = \frac{(1-q/p_k)^2}{1-p_k} n_k p_k \approx n_k p_k . \tag{D.1}$$

*This simplifies the first condition of Theorem 1 to a simple connectivity requirement. Hence, we can rewrite the conditions of Theorems 1, 2 as*

$$1 : n_k p_k \gtrsim \log n_k , \widetilde{D}(p_{\min}, q) \gtrsim \frac{\log n_{\min}}{n_{\min}} , \rho_{\min}^2 \gtrsim \max\{\sigma_{\max}^2, nq(1-q), \log n\} , \sum_{k=1}^{r} n_k^{-\alpha} = o(1) \text{ for some } \alpha > 0$$

$$2 : n_k p_k \gtrsim \log n , \widetilde{D}(p_{\min}, q) \gtrsim \frac{\log n}{n_{\min}} , \rho_{\min}^2 \gtrsim \max\{\sigma_{\max}^2, nq(1-q)\} .$$

**Example 1:** In a configuration with two communities $(n - \sqrt{n}, n^{-2/3}, 1)$ and $(\sqrt{n}, \frac{1}{\log n}, 1)$ with $q = n^{-2/3 - 0.01}$, we have $n_{\min} = \sqrt{n}$ and $p_{\min} = n^{-2/3}$. We have,

$$\widetilde{D}(p_{\min}, q) \approx n^{-2/3 + 0.01}$$

which does not exceed either $\frac{\log n_{\min}}{n_{\min}} \approx \frac{\log n}{\sqrt{n}}$ or $\frac{\log n}{n_{\min}} \approx \frac{\log n}{\sqrt{n}}$, and we get no recovery guarantee from Theorems 1 and 2 respectively. However, as $p_{\min} - q$ is not much smaller than $q$, while $\rho_{\min} \approx n^{1/3}$ grows much faster than $\log n$, the condition of Theorem 3 trivially holds.

Here are the related quantities for this configuration:

$$\rho_1 = n_1(p_1 - q) = (n - \sqrt{n})(n^{-2/3} - n^{-2/3 - 0.01}) \approx n^{1/3} \quad , \quad \rho_2 = n_2(p_2 - q) = \sqrt{n}(\frac{1}{\log n} - n^{-2/3 - 0.01}) \approx \frac{\sqrt{n}}{\log n}$$

which gives $\rho_{\min} \approx n^{1/3}$. Furthermore,

$$\sigma_1^2 = n_1 p_1 (1 - p_1) \approx n^{1/3} \quad , \quad \sigma_2^2 = n_2 p_2 (1 - p_2) = \frac{\sqrt{n}}{\log n} ,$$

which gives $\sigma_{\max} = \frac{\sqrt{n}}{\log n}$. On the other hand $nq(1-q) \approx n^{1/3 - 0.01}$ which is smaller than $\sigma_{\max}^2$.

**Example 2:** Consider a configurations with $(n - n^{2/3}, n^{-1/3 + \epsilon}, 1)$ and $(\sqrt{n}, \frac{c}{\log n}, n^{1/6})$ and $q = n^{-2/3 + 3\epsilon}$. Since all $p_k$'s and $q/p_k$'s are much less than $1$, the first condition of both Theorems 1 and 2 can be verified by Remark 2. Moreover, $n_{\min} = \sqrt{n}$ and $p_{\min} = n^{-1/3 + \epsilon}$ which gives

$$\widetilde{D}(p_{\min}, q) = n^{-\epsilon}$$

and verifies $\widetilde{D}(p_{\min}, q) \gtrsim \frac{\log n_{\min}}{n_{\min}}$ for 1, as well as $\widetilde{D}(p_{\min}, q) \gtrsim \frac{\log n}{n_{\min}}$ for 2. Moreover, $\rho_1 \approx n^{2/3+\epsilon}$ and $\rho_2 \approx \frac{\sqrt{n}}{\log n}$ which gives $\rho_{\min} \approx \frac{\sqrt{n}}{\log n} \gtrsim \sqrt{\log n}$. On the other hand, $\sigma_1^2 \approx n^{2/3+\epsilon}$ and $\sigma_2^2 \approx \sqrt{n}/\log n$ which gives

$$\max\{\sigma_{\max}^2, \, nq(1-q)\} \approx n^{2/3+\epsilon}.$$

Thus all conditions of Theorems 1 and 2 are satisfied. Moreover, as $p_{\min} - q$ is not much smaller than $q$, while $\rho_{\min} \approx \frac{\sqrt{n}}{\log n}$ is growing much faster than $\log n$, the condition of Theorem 3 trivially holds.

**Example 3:** Consider a configurations with $(\sqrt{\log n}, O(1), m)$ and $(n_2, O(\frac{\log n}{\sqrt{n}}), \sqrt{n})$ and $q = O(\log n/n)$, where $n_2 = \sqrt{n} - m\sqrt{\log n/n}$. Here, we assume $m \leq n/(2\sqrt{\log n})$ which implies $n_2 \geq \sqrt{n}/2$. Since all $p_k$'s and $q/p_k$'s are much less than 1, we can use Remark 2: the first condition of Theorem 1 holds as $n_1 p_1 \approx \sqrt{\log n} \gtrsim \log n_1 \approx \log \log n$ and $n_2 p_2 \approx \log n \gtrsim \log n_2$. However, $n_1 p_1 \approx \sqrt{\log n} \not\gtrsim \log n$ and Theorem 2 does not offer a guarantee for this configuration.

Moreover, $n_{\min} = \sqrt{\log n}$ and $p_{\min} = O(\frac{\log n}{\sqrt{n}})$ which gives

$$\widetilde{D}(p_{\min}, q) = \log n$$

and verifies $\widetilde{D}(p_{\min}, q) \gtrsim \frac{\log n_{\min}}{n_{\min}} \approx \frac{\log \log n}{\sqrt{\log n}}$ for 1, as well as $\widetilde{D}(p_{\min}, q) \gtrsim \frac{\log n}{n_{\min}} = \sqrt{\log n}$ for 2. Moreover, $\sigma_1^2 = \sqrt{\log n}$ (also $\rho_1$) and $\sigma_2^2 = \log n$ (also $\rho_2$) which gives

$$\max\{\sigma_{\max}^2, \, nq(1-q)\} \approx \log n$$

and $\rho_{\min}^2 \approx \log n$. For the last condition of Theorem 1 we need

$$m(\log n)^{-\alpha/2} + \sqrt{n}(\sqrt{n} - k\sqrt{\tfrac{\log n}{n}})^{-\alpha} = o(1)$$

for some $\alpha > 0$ which can be guaranteed provided that $m$ grows at most polylogarithmically in $n$. All in all, we verified the conditions of Theorem 1 while the first condition of 2 fails. Observe that $\rho_{\min}$ fails the condition of Theorem 3.

Alternatively, consider a configuration with $(\sqrt{\log n}, O(1), m)$ and $(\sqrt{n}, O(\frac{\log n}{\sqrt{n}}), m')$ and $q = O(\frac{\log n}{n})$, where $m' = \sqrt{n} - m\sqrt{\log n/n}$ to ensure a total of $n$ vertices. Here, we assume $m \leq n/(2\sqrt{\log n})$ which implies $m' \geq \sqrt{n}/2$. Similarly, all conditions of Theorem 1 can be verified provided that $m$ grows at most polylogarithmically in $n$. Moreover, the conditions of Theorems 2 and 3 fail to satisfy.

**Example 4:** Consider a configuration with $(\frac{1}{2}n^\epsilon, O(1), n^{1-\epsilon})$ and $(\frac{1}{2}n, n^{-\alpha}\log n, 1)$ and $q = n^{-\beta}\log n$, where $0 < \alpha < \beta < 1$ and $0 < \epsilon < 1$.

We have $\rho_1 \approx n^\epsilon$ and $\rho_2 \approx n^{1-\alpha}\log n$. Since $\rho_{\min}^2 \gtrsim \log n$, the last condition of Theorem 1 holds, and $\log n_{\min} \approx \log n$, we need to check for similar conditions to be able to use Theorems 1 and 2. Using Remark 2, the first condition of both Theorems holds because of $n_1 p_1 \approx n^\epsilon \gtrsim \log n$ and $n_2 p_2 \approx n^{1-\alpha}\log n \gtrsim \log n$. Moreover, the condition

$$\widetilde{D}(p_{\min}, q) \approx n^{\beta - 2\alpha}\log n \gtrsim \frac{\log n}{n_{\min}} \approx \frac{\log n}{n^\epsilon}$$

is equivalent to $\beta + \epsilon > 2\alpha$. Furthermore, $\sigma_1^2 = n^\epsilon$ and $\sigma_2^2 = n^{1-\alpha}\log n$, and for the last condition we need

$$\min\{n^{2\epsilon}, \, n^{2-2\alpha}\log^2 n\} \gtrsim \max\{n^\epsilon, \, n^{1-\alpha}\log n, \, n^{1-\beta}\log n\}$$

which is equivalent to $2\epsilon + \alpha > 1$ and $\epsilon + 2\alpha < 2$. Notice that $\beta + 1 > 2\alpha$ is automatically satisfied when we have $\beta + \epsilon > 2\alpha$ from the previous part.

**Example 5:** Consider a configuration with $(\log n, O(1), \frac{n}{\log n} - m\sqrt{\frac{n}{\log n}})$ and $(\sqrt{n\log n}, O(\sqrt{\frac{\log n}{n}}), m)$ and $q = O(\frac{\log n}{n})$. All of $\rho_1$, $\rho_2$, $\sigma_1^2$, $\sigma_2^2$, and $nq(1-q)$, are

approximately equal to $\log n$. Thus, the first and third conditions of Theorems 1 and 2 are satisfied. Moreover,

$$\widetilde{D}(p_{\min}, q) \approx 1 \gtrsim \frac{\log n_{\min}}{n_{\min}} \approx \frac{\log \log n}{\log n}$$

which establishes the conditions of Theorem 2. On the other hand, the last condition of Theorem 1 is not satisfied as one cannot find a constant value $\alpha > 0$ for which

$$\sum_{k=1}^{r} n_k^{\alpha} = \left( \frac{n}{\log n} - m\sqrt{\frac{n}{\log n}} \right) \log^{-\alpha} n + m(n \log n)^{-\alpha/2}$$

is $o(1)$ while $n$ grows.

**Example 6:** For the first configuration, Theorem 1 requires $f^2(n) \gtrsim \max\{\frac{\log n_1}{n_1}, \frac{\log n_{\min}}{n_{\min}}, \frac{n}{n_1^2}\}$ while Theorem 2 requires $f^2(n) \gtrsim \max\{\frac{\log n_1}{n_1}, \frac{\log n}{n_{\min}}, \frac{n}{n_1^2}\}$ and both require $n_{\min} \gtrsim \sqrt{n}$. Therefore, both set of requirements can be written as

$$f^2(n) \gtrsim \max\{\frac{\log n}{n_{\min}}, \frac{n}{n_1^2}\} \quad , \quad n_{\min} \gtrsim \sqrt{n}.$$

# E Statistical and Computational Regimes; A Literature Review

What we can infer about the community structure from a single draw of the random graph varies based on the regime of model parameters. Often, the following community retrieval scenarios are considered.

1. **Recovery,** where the proportion of misclassified nodes is negligible; either $0$ or asymptotically $0$ as the number of nodes grow, corresponding to the subregimes below.

   1a) *Exact Recovery (or Recovery with Strong Consistency).* In this regime it is possible to recover all labels, with high probability. That is, an algorithm has been proved to do so, whether in polynomial time or not. For example, [15, 3] studied the exact recovery problem for special cases of SBM.

   1b) *Almost Exact Recovery (or Recovery with Weak Consistency).* In this case, algorithms exist to recover a proportion $1 - o(1)$ of the nodes, but not all of them. See [34] for early works on weakly consistent recovery, [33] for the case of binary SBM, [40] for finite number of linear-sized communities, and [41, 19] for a growing number of approximately same-sized communities.

2. **Approximation,** where a finite fraction (bounded away from 1) of the vertices is recovered.

   2a) *Partial Recovery (or Approximation) Regime.* Only a *fraction* of vertices, i.e. $(1 - \alpha)n$ for some $0 < \alpha < 1$, can be guaranteed to be recovered correctly. This regime was first introduced in [16, 18]. A series of works have provided partial recovery conditions for the cases of two equivalent communities [32, 26, 30, 31], finite number of linear-sized communities [3], and heterogenous SBM [20, 24].

   2b) *Detectability.* One may construct a partition of the graph which is correlated with the true partition (which in this context means doing better than guessing), but one cannot *guarantee* any kind of quantitative improvement over random guessing. This happens in very sparse regimes when some $p_k$'s and $q$ are of the same, small, order; e.g. see [32, 4].

The levels of correct labeling in community detection described above can be studied from two points of view:

**Statistically,** one can ask about the parameter regimes for which the model can be retrieved based on one of the above objectives for retrieval (recovery or approximation).Such characterizations are specially important when an information-theoretical lower bound (below which retrieval is not possible with high probability) is shown to be achievable with an algorithm (with high probability), hence characterizing a *phase transition* in model parameters. Recently, there has been significant interest in identifying such *sharp thresholds* for various parameter regimes; e.g. [3] for the exact recoverability of a fixed number of linear-sized communities, [30, 26] (building upon [18, 32]) for detectability in binary SBM, and [4, 8] for more than two equivalent communities.

**Computationally,** one might be interested to study algorithms for recovery or approximation. In the older approach, algorithms were studied to provide upper bounds on the parameter regimes for recovery or approximation. See [13] or [3, Section 5] for a summary of such results. More recently, the paradigm has shifted towards understanding the limitations and strengths of tractable methods (e.g. see [29] on semidefinite programming based methods) and assessing whether successful retrieval can be achieved by tractable algorithms at the sharp statistical thresholds or there is a *gap*. So far, it is understood that there is no such gap in the case of exact recovery of binary SBM (e.g. via spectral methods in [33] or a partial recovery algorithm combined with a local improvement procedure in [2]), almost exact recovery of binary SBM (e.g., via spectral methods in [33, 40]), approximation of binary SBM (e.g., via weighted non-backtracking walks between vertices in [30] or counting self-avoiding walks in [26]), and exact recovery of linear-sized communities in [3]. However, this is still an open question for more general cases; e.g., see [4] and the list of unresolved conjectures therein.

As mentioned before, for SBM with only two equivalent communities, all of the above questions have been addressed in a series of recent papers [18, 32, 30, 26, 31, 33, 2, 21]. Apart from the binary SBM, the best understood cases are where there is a finite number $r$ of equivalent or linear-sized communities. As noted in [1], outside of the settings described above, the full picture has not yet emerged; many questions are unresolved, and many of the existing results give bounds that incorporate large or unknown constants.

## Footnotes

[1]As a more general result about the norms of rectangular matrices, but with the slightly stronger growth condition $\sigma^2 \geq \log^{6+\epsilon} n/n$.