[Reviews · NeurIPS 2016]

Reviewer 1

Summary

The authors provide results of exact recovery in the stochastic block model for general regimes (of number of clusters and their sizes).

Qualitative Assessment

The paper provides recovery guarantees for new (and quite general) regimes of the Stochastic Block Model. While the results are very general, they seem to be up to constants, which I believe is an important drawback. One issue is that there has been so much recent work on the Stochastic Block Model that it is very difficult to know exactly what is the state of the art (as it changes extremely rapidly) and very difficult to cite all relevant references (and indeed there are many missing, but this is not the author’s fault as it would impossible to include all) and so it is very difficult for me to identify exactly what are the contributions of this paper. One thing worth mentioning is that what the authors call ``partial observations’’ appears in the literature as ``Censored Block Model’’.

Confidence in this Review

1-Less confident (might not have understood significant parts)


Reviewer 2

Summary

The paper provides exact recoverability bounds for finding communities in heterogeneous stochastic block models. The paper uses semi-definite program (SDP) relaxations and maximum likelihood methods to give algorithms and then prove the existence of recovery of communities above certain thresholds for such algorithms. The paper also sufficient conditions for failure of exact recovery below the threshold.

Qualitative Assessment

The paper provides a series of results regarding the recovery of communities for heterogeneous stochastic block models. The theoretical justification for the results are also provided in the supplement. The paper provides quite explicit conditions for recovery of communities for both SDP relaxation algorithm and maximum likelihood algorithm. The paper also gives sufficient condition for failure of exact recovery for heterogeneous block models. The paper provides several nice examples for the recovery cases too. But, it would have been good if some simulation studies were made, specifically for the block models arising from the parameter space in between the recovery thresholds and sufficient condition for non-exact recovery. The part on the missing edges seem to be a bit tangential to the flow of the paper and it seems it can be addressed in a separate paper. The paper provides very nice theoretical results regarding heterogeneous block models. The results on SDP relaxation algorithms can become highly useful for practitioners too. The SDP relaxation algorithms for community detection are becoming quite popular recently but they are still notoriously slow. It would have been good to know, if any experiments were done with the convex relaxations and what size graphs were considered for the experiments. The paper is also quite well-written and the main points of the paper has been quite lucidly made.

Confidence in this Review

2-Confident (read it all; understood it all reasonably well)


Reviewer 3

Summary

The paper focuses on a heterogeneous extension of the well-known stochastic block models, by considering the case where the communities might vary significantly in both size and connectivity. The authors develop sufficient recovery conditions for both convex relaxation approaches and a modified maximum likelihood estimate, as well as some general lower bounds. A few concrete examples have been provided to illustrate their results. In particular, the paper is able to accommodate the scenario where some communities have vanishingly small size. The results are interesting, which provide a practically relevant extension of the prior work. In particular, I like the examples listed in Section 3, which help the readers develop a better understanding of the tradeoff in heterogeneous SBM. The flip side: the paper falls short of providing explanation for the key conditions presented in their main theorems (e.g. Theorem 1). While both upper and lower bounds on the performance of MLE have been presented, I couldn't find discussion concerning the gap between these bounds, making it difficult for me to evaluate the tightness of these results. Also, I think the paper would benefit from some numerical experiments.

Qualitative Assessment

1. It would be good to provide some operational meaning of the notion "relative density". 2. Some interpretation about the sufficient recovery condition in Theorem 1 is missing. What does the recovery condition mean? For example, the first condition might be interpreted as the mean difference (or signal strength) being larger than the variation. In general, I hope that the authors can try their best to provide intuition for each of the key conditions, which would be very helpful for the readers. 3. Around Equation (2.5), the authors turn their attention from the original maximum likelihood estimate to a modified MLE independent of the parameters. While this modified version is perhaps a more practically appealing algorithm, I'm wondering whether there is any statistical loss that occurs due to this modification. It would be good to provide some comparisons (either positive or negative). 4. The lower bound in Theorem 4 also needs some interpretation. At first glance it looks quite different from the sufficient recovery condition given in Theorem 3. The authors might want to discuss the tightness of both Theorem 3 and Theorem 4 -- do they match at least for some special cases? 5. While the paper is mostly concerned with the scaling results, can the authors comment on the possibility of obtaining tight pre-constants? 6. Page 5, the expression "sufficient conditions for when the exact recovery is impossible" doesn't read well; perhaps just "necessary conditions for exact recovery". 7. One of the main results is the power of convex relaxation even when some communities have very small sizes. Can the authors provide some numerical experiments using convex programming to illustrate the practical relevance of the theoretical prediction?

Confidence in this Review

2-Confident (read it all; understood it all reasonably well)


Reviewer 4

Summary

This paper discussed the conditions of recoverability of the heterogenous stochastic block model under more general settings, where there is no restriction on the number and sizes of communities or how they grow with the number of nodes, as well as on the connectivity probabilities inside or across communities. The author(s) provided results from both statistical and computational aspects of recoverability. The examples in the paper offered a better understanding of the abstract theorems. The author(s) also identified some parameter configurations that can be efficiently recoverable by semidefinite programs. This can provide some guidance in practice of the usage of semidefinite programming.

Qualitative Assessment

The problem and results are clearly stated, easy to follow. 1. Section 2.3 Partial Observations. The authors assumed that "the entries of the A (adjacency matrix) has been observed independently". Would this assumption be too strong in practice? It might be better if the authors could add some discussion about how the conclusions would change if this assumption was violated. 2. In Table 1, Ex. 1 and Ex. 2 have the same importance, but different recoverability by Thm. 1 and Thm. 2. It would be better if the authors could list the differences of these two example in the table as well.

Confidence in this Review

1-Less confident (might not have understood significant parts)


Reviewer 5

Summary

The authors study the problem of recovering communities under the stochastic block model and show that semidefinite programming can recover communities with different characteristics (e.g. smaller) than previous analyses.

Qualitative Assessment

I enjoyed reading this paper: * Paper is well organized and easy to follow. * The possibility of recovering communities smaller than sqrt(log n) based on convex optimization is an appealing theoretical result. * The paper abounds with examples that make the theoretical results easier to interpret. Table 1 is very helpful to identify the theorem that is applicable on different regimes. Theorem 1 (and 2) assures that under some regularity conditions \hat{Y} and Y^* coincide. However, \hat{Y} may can be continuous and Y^* is a binary matrix. Isn't there projection or thresholding step at the end of problem? It is not clear under which set the maximizationof (2.4) is carried out, please specify.

Confidence in this Review

1-Less confident (might not have understood significant parts)


Reviewer 6

Summary

The paper is well written and with some novel results on community detection for the case of the so-called "heterogeneous SBM". Strong Points: 1) The results are novel to the best of my knowledge and to a regime (heterogeneous) that no significant prior work exists. Weak Points: 1) The optimization problems used are not that novel. The obtained results seem to be an "analysis artifact" only due to stronger matrix concentration results used by the authors than in pre-existing literature. 2) The topic is not that novel, meaning that a lot of work has been published on it the last decade. 3) The proof techniques used are very standard. Overall, I liked reading this paper and I think that the heterogeneous regime may worth more investigation.

Qualitative Assessment

Based on the submitted file, which the visible one, I would like to see numerical demonstration of the theoretically derived results. Moreover, I think that the paper might fit well for a poster presentation since the topic is hot and the explored regime allows for more investigation.

Confidence in this Review

2-Confident (read it all; understood it all reasonably well)